



# Radiation fog formation alerts using attenuated backscatter power from automatic Lidars and ceilometers

Martial Haeffelin[1], Quentin Laffineur[2], Juan-Antonio Bravo-Aranda[1], Marc-Antoine Drouin[3], Juan-Andrés Casquero-Vera[3,4], Jean-Charles Dupont[5], Hugo De Backer[2]

[1]Institut Pierre Simon Laplace, Ecole Polytechnique, Centre National de la Recherche Scientifique, Palaiseau, 91128 France
[2]Royal Meteorological Institute of Belgium, Uccle, 1180, Belgium
[3]Laboratoire de Météorologie Dynamique, Ecole Polytechnique, Centre National de la Recherche Scientifique, Palaiseau, 91128 France
[4]Department of Applied Physic, University of Granada, Granada, 18071, Spain
[5]Institut Pierre Simon Laplace, Université Versailles Saint Quentin-en-Yvelines, Guyancourt, 78280, France

*Correspondence to*: Martial Haeffelin (martial.haeffelin@ipsl.polytechnique.fr)

**Abstract.**

Radiation fog occurs over many locations around the world in stable atmospheric conditions. Air traffic at busy airports can be significantly disrupted because low visibility at the ground makes it unsafe to take off, land and taxi on the ground. Current numerical weather prediction forecasts are able to predict general conditions favorable for fog formation, but not the exact time or location of fog occurrence. A selected set of observations available in near realtime at strategic locations could also be useful to track the evolution of key processes and key parameters that drive fog formation. Such observations could complement the information predicted by NWP models that is made available to airport forecasters in support of their fog forecast. This paper presents an experimental setup based on collocated automatic lidar and ceilometer measurements, relative humidity measurements and horizontal visibility measurements to study hygroscopic growth of fog condensation nuclei. This process can take several minutes to hours and can be tracked using lidar or ceilometer attenuated backscatter profiles. Based on hygroscopic growth laws we derive a set of parameters that can be used to provide alerts minutes to hours prior to formation of radiation fog. We present an algorithm that uses the temporal evolution of attenuated backscatter measurements to derive pre-fog formation alerts. The performance of the algorithm is tested on 45 independent pre-fog situations at two locations (near Paris, France and Brussels, Belgium). We find that pre-fog alerts occur predominantly 10-50 min prior to fog formation at an altitude ranging 0 to 100 m above ground. In a few cases, alerts can occur up to 100 min



prior to fog formation. Alert durations are found to be sensitive to relative humidity conditions found a few hours prior to the

fog.



## 1. Introduction

Radiation fog occurs over many locations around the world in stable atmospheric conditions when cooling provides supersaturated conditions allowing persistent droplet activation and formation of a liquid layer near the ground. When this occurs at busy airports air traffic can be significantly disrupted because low visibility at the ground makes it unsafe to take

off, land and taxi on the ground. So in the case of low visibility at the ground, the rate of take off and landing must be significantly reduced possibly affecting flying schedules across an entire country or continent. Hence accurate fog formation and dissipation forecasting could significantly help improve flight planning and reduce fuel usage. Increasing air traffic in the past decade could potentially explain the renewed interest in fog forecasting at airports worldwide, such as in Paris, Belgrade, Melbourne, Kolkata, or Cape Town (Bergot et al. 2015; Veljovic et al. 2015; Boneh et al. 2015; Dutta and

Chaudhuri 2015; Van Schalkwyk and Dyson 2013). In some locations, fog occurs only a few times per year, yet since it occurs unexpectedly, disruptions and hazards may reach unacceptable levels.

Current numerical weather prediction forecasts are able to predict general conditions favorable for fog formation (e.g. high relative humidity, radiative cooling conditions, moderate mixing), but not the exact time or location of fog occurrence (e.g. Steeneveld et al. 2015; Roman-Cascon et al. 2015). Such models lack the vertical and spatial resolution and representation of

boundary layer and microphysical processes to accurately represent actual near-surface cooling rates, turbulent mixing, supersaturation, and their vertical structure in the surface layer. They typically do not represent accurately the activation processes of fog droplets that depend on the chemical nature of the aerosols, on their size distributions, and on typically very low supersaturation conditions. Higher-resolution models (i.e. large eddy simulation models) can be used to improve the representation of fine scale dynamical and thermodynamical processes, however microphysical and chemical processes still

must be represented using parametrizations. Roman-Cascon et al. (2015) provide a detailed overview of recent studies investigating new technical configurations or physical parametrizations to improve fog forecasting scores of numerical weather prediction models.

A selected set of observations available in near realtime at strategic locations could also be useful to track the evolution of key processes and key parameters that drive fog formation. Such observations could complement the information predicted

by NWP models by providing true values of key parameters that could make the difference between an imminent formation





(or not) of fog and likewise for dissipation. Direct use of satellite or ground-based observations is already commonly used for short-term precipitation estimates (e.g. Ebert et al., 2007). Statistical methods using measured time series of key parameters are also used for wind or solar power production forecasting.

To illustrate this, let us take the example of radiation fog formation that is characterized by the formation of liquid water in

the atmosphere. The phase change from water vapor to liquid water yields a rapid increase in the optical cross-sections of scattering particles that are responsible for the rapid decrease in visibility. This phase change is preceded by progressive hygroscopic growth of fog condensation nuclei, a process that can take minutes to hours depending on the cooling rate leading to supersaturated conditions. Hygroscopic growth of aerosols has been studied extensively, and physical laws have been established empirically between relative humidity and aerosol optical parameters. Such fog formation precursor

processes could be tracked with proper instrumentation. The activation process resulting from cooling can occur at the surface or aloft. Hence it is important to be able to track this process over a sufficiently deep vertical profile to capture activation where it occurs first.

Here we apply hygroscopic growth laws to establish a method that uses attenuated backscatter measured by automatic lidars and ceilometers (ALCs) to track activation of aerosols into fog or low-cloud droplets. As most airports are equipped with

ALCs, near realtime analysis of attenuated backscatter measurements could provide useful warnings to airport forecasters about when radiation fog is likely or not likely to occur. In this paper, we study relationships between aerosol properties, relative humidity, and light scattering measured by ALCs (Section 3). We present a method and experimental setup to investigate aerosol hygroscopic growth by means of attenuated backscatter profiles and a meteorological station on a tower (Section 3). We present a methodology to use attenuated backscatter measurements to derive pre-fog formation alerts

(Section 4). Finally we discuss the occurrences and characteristics of these alerts based on 45 case studies (Section 5).

## 2. Sites, instrumental issues and datasets

### 2.1. Sites and instruments used in this study

This study relies on measurements from two locations, near Paris, France, and Brussels, Belgium. The first site is the SIRTA atmospheric research observatory located in Palaiseau (2.208˚ E, 48.713˚ N, 160 m ASL), ~ 20 km south of Paris city center



[*Haeffelin et al.*, 2005]. SIRTA is situated on the plateau of Saclay in a suburban environment in the flat Parisian plain, surrounded by small suburban towns, agricultural fields, forests and major roads connecting the suburban areas with the city of Paris. A large suite of state-of-the-art active and passive remote sensing instruments and in-situ sensors are operated at SIRTA since 2002. These instruments are used to document atmospheric state variables and relevant meteorological and

climatological parameters. SIRTA has hosted major national and international field campaigns, in particular the ParisFog campaign series from 2006 to 2014, during which numerous instruments were gathered each winter (October to March) to monitor dynamical, thermodynamical, radiative, optical, microphysical properties and processes that drive formation, development and dissipation of fog (see Haeffelin et al., 2010; Hammer et al. , 2014; Dupont et al., 2015).

Several instruments deployed at SIRTA are of interest to this study, namely: a Vaisala CL31 ALC (see main characteristics

in Table 1) providing profiles of attenuated backscatter at 905 nm, visibility sensors (Degreane DF20/DF20+) providing visibility at 4 and 20 m agl, and a tower with in-situ sensors providing temperature and relative humidity measurements at 1, 2, 5, 10, 20 and 30 m agl. All measurements are synchronized and processed on the same 1-min temporal grid.

The second site is the Plateau of Uccle located at Uccle, Belgium (4.350° E, 50.800° N, 100m ASL), ~ 5 km south of Brussels city center. It is situated in a residential suburb of Brussels surrounded by woods, grasslands, suburban and urban

environments. Three institutions (IRM-KMI, ORB-ROB and ISAB-BIRA) on the Plateau of Uccle manage on the long-term active and passive remote sensing atmospheric instruments and in-situ sensors. Three of these instruments are used in this study: a Vaisala CL51 ALC (see main characteristics in Table 1) providing profiles of attenuated backscatter at 905 nm computed with a resolution of 1 min, a webcam providing horizon images with a sampling rate of 5 min and in-situ sensors providing temperature and relative humidity measurements at 2 m AGL computed with a resolution of 10 min. This latter

dataset was oversampled at 1 min by repeating the same measurements over each computed interval of 10 min and is synchronized with the ALC measurements.

### 2.2.  Automatic Lidars and Ceilometers

Automatic single wavelength backscattering Lidars and profiling Ceilometers (ALCs) operate at ultraviolet (355 nm), visible (532 nm) or near-infrared wavelengths (880, 905, 1064 nm). They provide a vertically resolved signal (with variable vertical

resolution on the order of 1-10 m) that is proportional to the scattering cross section of molecules (Rayleigh scattering) and



particles (Mie scattering) whose size range from nanometers to micrometers. ALCs are characterized by robust designs allowing continuous unattended operation and low emission power to ensure eye-safety. They are also characterized by low signal-to-noise values, in particular during daytime and at high altitude. ALCs are available from several manufacturers. Several hundred such ALCs are currently operated in Europe (e.g. Haeffelin et al. 2012; DWD ceilometer webpage[1]). The E-

PROFILE program is setting up a framework to centralize and harmonize processing and distribution of EU ALC measurements[2].

Although these instruments are considered as general-purpose cloud height sensors, their aerosol profiling capabilities are also recognized (e.g. Markowicz et al., 2008). Absolute calibration of ALC backscatter profiles can be achieved using different techniques (e.g. O'Connor et al., 2004; Wiegner and Geiss, 2012). Then particle backscatter coefficients can be

retrieved with relative errors on the order of 10% or less (Wiegner and Geiss, 2012). Madonna et al. (2015) point out however that ALC calibration can also be affected by variability in external temperature, in water vapor, optical overlap functions, signal-to-noise ratio and dynamic range. So ALC data must be used with caution when retrieving quantitative aerosol optical properties.

ALC could be particularly suited for applications where absolute calibration is not needed. For example, tracking the depth

of the atmospheric layer over which surface emitted species are mixed (so called mixing layer) can be done by searching for gradients in the vertical profile of ALC measurements (e.g. Morille et al., 2007; Baars et al., 2008). Determination of gradients does not require prior absolute calibration of the signal. However, understanding how aerosol backscatter profiles can be used to track vertical mixing processes may require more information than just the ALC profile (e.g. Haeffelin et al., 2012; Pal et al. 2013).

A key issue for low-altitude (0 -1000m) profiling applications is the optical overlap between the emitted laser pulses and the collecting optical field. The optical overlap function ranges between 0 and 1. It is 0 at the ground and reaches 1 at the altitude where all backscattered photons can be accounted for by the collecting optical field. For mono-axial ALCs full overlap can be reached at low altitudes (i.e. 20-200m), while for bi-axial ALCs full overlap can be reached at higher altitudes

---

[1] http://www.dwd.de/DE/forschung/projekte/ceilomap/ceilomap_node.html

[2] http://www.eumetnet.eu/alc-network





(i.e. 500-1500 m). The CL31 and CL51 measurements used in this study have been corrected for incomplete optical overlap by the manufacturer. For our application, CL31 (CL51) data can be exploited starting at 10 m (50 m) agl.

### 2.3. Fog dataset description

Glickman (2000) provides a commonly accepted definition of fog as "water droplets suspended in the atmosphere in the

vicinity of the Earth's surface that reduce visibility below 1 km". In our study, we focus on radiation fog, i.e. fogs that form in conditions characterized by near-surface radiative cooling, decreasing temperature and/or increasing relative humidity near the surface, predominantly cloud-free atmosphere. Three distinct types of situations are of interest: (1) vertically developed radiation fog (D-RFOG), (2) shallow radiation fog (S-RFOG), (3) quasi radiation fog (Q-RFOG). Following the definition of Dupont et al. (2015), a vertically developed radiation fog exceeds 20 m in depth. It is most frequently 100-

500m deep. A shallow radiation fog is less than 20 m deep. Its depth ranges from 2 to 20 m. The horizontal visibility in the fog layer is less than 1 km for both types of radiation fogs. In a quasi radiation fog, the horizontal visibility ranges between 2 and 3 km (Dupont et al. 2015). The low visibility is due to inactivated haze particles (Kokkola et al. 2003). QFOG atmospheric conditions are similar to conditions encountered ahead of a shallow or a deep RFOG (described above). Our study relies on 45 cases of deep radiation fog observed at SIRTA and UCCLE.

### 3. Relationships between aerosol, humidity, and light scattering

When relative humidity (RH) increases in the atmosphere, aerosol particles may swell due to water uptake becoming significantly larger than in dry conditions. This process, called hygroscopic growth, is typically continuous and smooth without sudden increase or reduction of water absorption, except for pure salt particles whose growth is characterized by a

sharp increase at the critical humidity (Winkler, 1973). Atmospheric aerosols can be divided into deliquescent species and insoluble species according to their change in mass when humidified where their hygroscopic growth depends on their size and chemical composition (McMurry and Stolzenburg (1988); Svenningsson et al., 1992). Inorganic and sea salts, included in fine and coarse modes, respectively, show a high hygroscopicity whereas black carbon and organic matter (fine mode) and mineral dust (coarse mode) show a low or very low hygroscopicity (Zieger et al., 2013; Titos et al., 2014). However,

hygroscopicity can change due to coating processes (Semeniuk et al., 2007). For example, Hatch et al., (2008) indicate that a





small amount of coating by humic substances on insoluble mineral particles can enhance the water absorption. In a similar way, Zhang et al., (2008) points out that aged soot particles, coated by sulphuric acid, show a larger hygroscopic growth than fresh soot.

### 3.1. Relationship between aerosol particle size and RH

The relationship between aerosol particle size and RH has been deeply studied (e.g., Hänel, 1976; Pitchford and McMurry, 1994; Hämeri et al., 2000). Typically, this relationship is described by the aerosol diameter growth factor, $g(RH)$, defined as the wet-to-dry particle radius ratio ($r/r_d$). It is expressed as

$$g(RH) = \left(1 + \frac{\rho_d}{\rho_w}\mu(RH)\frac{RH}{1-RH}\right)^{1/3}$$
Eq. (1)

where RH has been expressed as a fraction, $\rho_d$ and $\rho_w$ are the dry particle and water density, respectively, and $\mu(RH)$ is the linear mass increase coefficient (Hänel, 1976).

Atmospheric RH alters the optical and microphysical properties of atmospheric aerosols due to aerosol hygroscopic growth. For example, the mass scattering efficiency of aerosol particles depends on their particle size and refractive index, the latter characterized by the chemical composition (Pilinis and Pandis, 1995). Conversely to the particle size increase, refractive index (both real and imaginary parts) decreases with increasing RH changing the lidar backscatter-to-extinction ratio. The real part of the water refractive index is lower than the aerosol one, leading to a decrease in scattering. The

imaginary part of water is zero and thus, a decrease in absorption can be expected. However, the refractive index decrease is not large enough to counteract the increase in particle cross-section that is proportional to the radius squared. Thus, the size dependence dominates, leading to an increase in scattering as RH increases.

Due to this phenomena long-term in situ measurements of aerosol optical and microphysical properties are usually performed at standardized dry conditions (WMO/GAW, 2003). Many investigators (e.g., Kotchenruther and Hobbs 1998;

Kotchenruther et al. 1999; Raut and Chazette 2007; Randriamiarisoa et al. 2006) have analysed the relationship between the aerosol light scattering coefficients, $\sigma_{sp}$, and RH using the semi-empirical equation developed by Kasten, (1969), and more rigorously discussed by Hänel, (1976),



$$\sigma_{sp}(RH) = \sigma_{sp}^{dry}(1 - RH)^{-\gamma} \qquad \text{Eq. (2)}$$

where $\sigma_{sp}^{dry}$ is the scattering coefficient of dry particles and $\gamma$ is an empirical fitting parameter. Then, aerosol properties can

be computed at any RH using the scattering hygroscopic enhancement factor, $f(RH)$, defined commonly as the ratio of $\sigma_{sp}$

at a given RH, $\sigma_{sp}(RH)$, to that in dry condition, $\sigma_{sp}^{dry}$,

$$f(RH) = \frac{\sigma_{sp}(RH)}{\sigma_{sp}^{dry}} \qquad \text{Eq. (3)}$$

Using Eq. (2), Eq. (3) can be rewritten as

$$f(RH) = \left(\frac{1 - RH}{1 - RH_{dry}}\right)^{-\gamma} \qquad \text{Eq. (4)}$$

for a reference value of RH, $RH_{dry}$, where no hygroscopic growth is expected. This is one of the physical parameters

commonly applied to describe aerosol hygroscopicity (Hänel, 1976; Zieger et al, 2013; Titos et al., 2014 and references

therein). According to Wulfmeyer and Feingold (2000), this function is mainly sensitive to the mass fraction of soluble

material of the internally mixed aerosol (also called solubility). It is worthy to note that $g(RH)$ and $f(RH)$ (Eq. (1) and Eq.

(4), respectively) are different since the particle shape affects the aerosol optical and hygroscopic properties, leading to

significant impact on $f(RH)$.

      Even though $f(RH)$ has been mainly derived from $\sigma_{sp}$, it has been applied to other optical and microphysical

properties as the aerosol-particle extinction coefficient and the volume concentration (Veselovskii et al., 2009) and the

aerosol-particle backscatter coefficient and fine-mode volume concentration (Granados-Muñoz et al., 2015) derived from

lidar measurements. In this regard, Granados-Muñoz et al., (2015) describe that the hygroscopic growth function is valid for

all aerosol properties and thus,

$$f_\zeta(RH) = \frac{\zeta(RH)}{\zeta(RH_{ref})} \qquad \text{Eq. (5)}$$

where $f_\zeta(RH)$ is the hygroscopic factor obtained for an aerosol property, $\zeta(RH)$, and $RH_{ref}$ is the reference RH for which

no hygroscopic growth is expected. These studies use the vertical variation of aerosol-particle properties within a well-mixed

layer where RH, measured by radiosonde, increase/decrease with height. The advantage of this technique, as opposed to



laboratory or most in situ studies, is that it can provide relatively continuous altitude-resolved measurements without

perturbing the aerosol or its surroundings. Additionally, RH ranges analyzed by in-situ measurements usually remain below

90-95% to prevent condensation on the instrumentation (Titos et al., 2014b). The obvious disadvantages are:

- the limited occasions for which the same aerosol type exists in a given portion of a profile that is characterized by

widely changing RH because the sampled aerosols are not controlled

- the requirements of collocated aerosol-particle properties (e.g., lidar) and RH (e.g., radiosonde) vertical profiles.

In this work, we propose a new experimental setup to track the aerosol-particle hygroscopic growth by means of an

ALC and a hygrometer installed in a tower. The theoretical basis and the experimental setup are explained in the next

section.

**3.2.  Aerosol-particle hygroscopic growth by ceilometer: theoretical basis and experimental setup**

The ALC signal is expressed by

$$P(t,r) = C(t) \cdot \frac{O(t,r)}{r^2} \cdot \beta(t,r) \cdot T^2(t,r) \qquad \text{Eq. (6)}$$

where $t$ is the time, $r$ is the range, $C$ represents the height-independent properties (e.g., the outgoing laser power, detection

efficiency, …), $O(t,r)$ is the overlap function, $\beta(t,r)$ is the aerosol backscatter coefficient and $T(t,r)$ is the transmittance.

Wavelength dependence has been omitted for simplicity. Defining the attenuated total backscatter coefficient, $\beta_{att}$, as $\beta \cdot$

$T^2$ and correcting the signal by range, the range corrected signal, $RCS$, can be written as follow,

$$RCS(t,r) = C(t) \cdot O(t,r) \cdot \beta_{att}(t,r) \qquad \text{Eq. (7)}$$

Since the $RCS$ and RH are continuously measured at the tower height, $r_t$, where the hygrometer is installed, we can track the

hygroscopic growth during a certain period dividing the $RCS(t,r_t)$ by a reference value $RCS(t_d,r_t)$ measured at a time

considered as dry,

$$\frac{RCS(t,r_t)}{RCS(t_d,r_t)} = \frac{C(t) \cdot O(t,r_t) \cdot \beta_{att}(t,r_t)}{C(t_d) \cdot O(t_d,r_t) \cdot \beta_{att}(t_d,r_t)} \qquad \text{Eq. (8)}$$

Since $C(t)$ and $O(t,r_t)$ variations can be neglected in periods of hours,





$$\frac{RCS(t,r_t)}{RCS(t_d,r_t)} \cong \frac{\beta_{att}(t,r_t)}{\beta_{att}(t_d,r_t)}$$ 

Eq. (9)

Then, defining the $\beta_{att}$ hygroscopic growth function as,

$$f_{\beta_{att}}(RH) = \frac{\beta_{att}(RH)}{\beta_{att}(RH_{ref})}$$ 

Eq. (10)

the ratio of Eq. (9) is a direct measurement of $f_{\beta_{att}}(RH)$. However, since both particles and molecules contribute to $\beta_{att}$, it is necessary to derive the relationship between $f_{\beta_{att}}(RH)$ and the particle backscatter coefficient, $f_{\beta_p}(RH)$. To this aim, we firstly show the relationship between $f_{\beta_{att}}(RH)$ and the total backscatter coefficient, $f_\beta(RH)$ substituting the $\beta_{att}$ definition

in Eq. (10),

$$f_{\beta_{att}}(RH) = \frac{\beta(RH)}{\beta(RH_{ref})}\frac{T^2(r,RH)}{T^2(r,RH_{ref})}$$ 

Eq. (11)

where the transmittance ratio can be rewritten as below,

$$f_{\beta_{att}}(RH) = f_\beta(RH)exp\left(-2\int_0^r \left(\alpha(r,RH) - \alpha(r,RH_{ref})\right)d\xi\right)$$ 

Eq. (12)

Assuming that the integration range is small enough to assume height-constant extinction, i.e., and using the extinction hygroscopic growth, $f_\alpha(RH)$ to rewrite $\alpha(r,RH)$ as $f_\alpha(RH)\alpha(r,RH_{ref})$, the relationship between $f_{\beta_{att}}(RH)$ and $f_\beta(RH)$ becomes,

$$f_{\beta_{att}}(RH) = f_\beta(RH)e^{-2\cdot r_{mat}\cdot\alpha(R,RH_{ref})\cdot(f_\alpha(RH)-1)}$$ 

Eq. (13)

In order to quantify the differences between $f_{\beta_{att}}(RH)$ and $f_\beta(RH)$, $f_{\beta_{att}}(RH)$ is simulated given values to $R$, $f_\beta(RH)$, $f_\alpha(RH)$ and $\alpha(R,RH_{ref})$: $r_{mat}$ is 30 m (hygrometer altitude), $f_\beta(RH)$ and $f_\alpha(RH)$ vary in the range [2,10] and [3,11] (assuming that $f_\beta(RH) < f_\alpha(RH)$ ) and $\alpha(r_{mat},RH_{ref})$ has been parameterized by varying the lidar ratio at $RH_{ref}$ in the range [30, 80] sr with $\beta(r_{mat},RH_{ref}) = 2\cdot10^{-6}\ m^{-1}sr^{-1}$. As it can be seen in Figure 1, $f_{\beta_{att}}(RH)$ and $f_\beta(RH)$ are very similar with differences less than 10%.

Next we study the relationship between the total, $f_\beta(RH)$, and particle, $f_{\beta_p}(RH)$, backscatter coefficient hygroscopic growth.

According to their definition, the hygroscopic growth of the total-to-particle backscatter coefficient ratio is expressed by,





$$\frac{f_\beta(RH)}{f_{\beta_p}(RH)} = \frac{\beta(RH)\Big/\beta\left(RH_{ref}\right)}{\beta_p(RH)\Big/\beta_p\left(RH_{ref}\right)}$$ Eq. (14)

Since $\beta = \beta_m + \beta_p$ and the backscattering ratio, R, is defined by $R = \beta/\beta_m$, the previous equation can be rewritten in terms of R as follow,

$$\frac{f_\beta(RH)}{f_{\beta_p}(RH)} = \frac{R(RH)}{R\left(RH_{ref}\right)} \frac{\left(R\left(RH_{ref}\right)-1\right)}{(R(RH)-1)}$$ Eq. (15)

Considering that the particle backscattering is much greater than the molecular one for large wavelengths (i.e., $R \gg 1$), we can neglect the difference between $f_\beta(RH)$ and $f_{\beta_p}(RH)$ since $R(RH) \approx (R(RH)-1)$. Figure 2 shows the relative

5    differences between $f_\beta(RH)$ and $f_{\beta_p}(RH)$ for different backscattering ratios parameterizing $R\left(RH_{ref}\right)$. As it can be seen, R values larger than 5 are enough to ensure a relative difference between $f_\beta(RH)$ and $f_{\beta_p}(RH)$ below 10%. Therefore, although $f_{\beta_{att}}(RH)$ is influenced by the particle and molecular contribution, it has been demonstrated that it can provide information about the aerosol-particle hygroscopic growth. Based on this theory, we used synergetic measurements of a CL31 ALC and a hygrometer located at 30 m in a measurement tower. Distance between the ALC and the tower is about 10

10    m. Figure 3 shows the $f_{\beta_{att}}(RH)$ as function of the relative humidity as well as its fitting to the Hänel function (see Eq. (4)). A $\gamma$ parameter of 0.64 is derived from fitting the data. The strong correlation between $f_{\beta_{att}}(RH)$ and the Hänel function ($R^2=0.99$) demonstrates the influence of the hygroscopic growth on $\beta_{att}$. Analyses of 5 other cases measured at SIRTA yield $\gamma$ parameters ranging from 0.24 to 0.64 (all cases have $R^2>0.9$ fitting of $f_{\beta_{att}}(RH)$).

### 4. Description of the Predictive Alert of Radiation FOG (PARAFOG) algorithm

As demonstrated in Sect. 3, the use of $\beta_{att}$ measurements from ALC in Eq. (10) can provide information about the aerosol-particle hygroscopic growth process. Real-time monitoring of this hygroscopic growth function can provide information on the dynamics of pre-fog aerosol activation processes in the atmosphere from the ground and up in the mixing layer. An algorithm analyzing the temporal evolution of ALC attenuated backscatter measurements in a 400-m deep layer above the

20    ground is presented here. This algorithm is intended to provide alerts that aerosol backscatter is changing at a fast rate, which



is typically observed when saturation conditions are reached and aerosols activate into cloud or fog droplets. This section presents how ALC measurements can be used to provide pre-fog alerts that could be used to support a decision making process regarding likelihood of radiation fog occurrence at the ground. Because fog formation, development and dissipation are influenced by multiple processes, analyzing ALC measurements should not be thought as a stand alone unique fog

prediction tool. Rather such realtime analysis should be used to more precisely identify when and where a critical process is occurring that is likely to lead to fog occurrence at the location where the measurement is made.

### 4.1. Critical steps of PARAFOG

Our algorithm development had two main principles in mind: (1) clearly identify when conditions are favorable and not favorable for pre-fog alerts, because fog is a rare event, hence the alert system should be OFF most of the time. (2) use parameters, analyses and tests that are as widely applicable as possible so that they can be applied to different instruments, locations and atmospheric conditions. Following these principles, the PARAFOG algorithm is based on 4 critical steps:

1. PARAFOG start: Determination if conditions are favorable for radiation fog formation and determination of
reference ALC attenuated backscatter profile.
2. Hygroscopic growth detection: Computation of the rate of change of the hygroscopic growth function.
3. Alert levels: Computation of pre-fog alert levels depending on the rate of change of hygroscopic growth and ALC attenuated backscatter.
4. End of PARAFOG: Determination if conditions are no longer favorable for radiation fog

***Step 1 (Start).*** Fog is a relatively rare event. PARAFOG must be activated in conditions that are potentially favorable for fog formation. PARAFOG must remain "OFF" otherwise. Two atmospheric parameters are used to activate PARAFOG: relative humidity that drives the hygroscopic growth of aerosol particles and sky conditions that drive the infrared radiative cooling of the surface layer. Both parameters play a key role in radiation fog formation. To turn PARAFOG "ON", the near-surface

relative humidity (typically measured at 2 m agl) must remain higher than 85% and cloud cover fraction must remain below





50% during at least 10 min. Both parameters could be adjusted according to geographical locations. Two cloud cover fractions are considered: (1) cloud cover ranging from 0 to 400 m agl, and (2) cloud cover ranging from 400 m to the top of the ALC range. Both cloud cover fractions are derived from the cloud base height measurement of the ALC.

The occurrence of low-level clouds or fog at an altitude less than 400 m agl when PARAFOG was already "ON" doesn't

5    turn off the algorithm. The 400-m limit is based on a study by Dupont et al. (2015) based on over 100 fog events that show that in conditions favorable for fog formation the moist surface layer is generally less than 300-m deep and that clouds that form at altitudes higher than 400 m very rarely subside all the way to the ground to form fog.

***Step 2 (Hygroscopic growth).*** When PARAFOG is turned "ON", the hygroscopic growth function for each altitude $r_j$ (with $j$ the level index) from 0 to 400 m (a.g.l.) at time $t_m$ is estimated using Eq. (10):

$$f_{\beta_{att}}\left(t_m, r_j, RH(t_m, r_i)\right) = \frac{\beta_{att}(t_m, r_j, RH(t_m, r_i))}{\beta_{att}(r_j, RH_{ref}(t_{ref}, r_i))} \qquad \text{Eq. (16)}$$

where $\beta_{att}\left(r_j, RH_{ref}(t_{ref}, r_i)\right)$ is derived when $RH(r_i)$ is minimum at the level $r_i$ ($i$ is level index where $RH$ is measured) closest to $r_j$ over the time interval $\Delta T$ before PARAFOG was turned on for the first time ($t_{on}$):

$$RH_{ref}\left(t_{ref}, r_i\right) = \min_{t_m - \Delta T \le t_m \le t_{on}} RH(t_m, r_i) \qquad \text{Eq. (17)}$$

with $i$ corresponding to:

$$\min_{0 < i \le n} \left| r_i - r_j \right|$$

15    with $n$ the number of $i$ levels. Identifying $t_{ref}$ is a critical step because the hygroscopic growth function is very sensitive to the reference $\beta_{att}$ profile (see discussion in Sect. 5).

A $\Delta T$ of 10 hours is used to reduce the risk to get a $RH_{ref}$ higher than 75% that may reduce the $f_{\beta_{att}}$ value due to high values of reference $\beta_{att}$ inducing potentially a delay in fog warning (see Sect. 5). Reference $\beta_{att}$ profiles that include clouds are rejected.

20    Figure 4 shows a simulation of the temporal evolution of $f_{\beta_{att}}\left(t_m, r_j, RH(t_m, r_i)\right)$ for one level by using Eq. (4) ($RH_{dry} = RH_{ref}$) with $\gamma = 0.7$ (Fig. 4b) and the temporal evolution of $\partial f_{\beta_{att}} / \partial t$ (Fig. 4a) for a situation where $RH$ changes at a




constant rate of 2% per hour (Fig. 4c), continuously for 10 hours. Figure 4b shows that the absolute value of $f_{\beta_{att}}$ is quite sensitive to $RH_{ref}$. Hence thresholds to identify significant hygroscopic growth required to trigger fog alert levels would not be widely applicable. Using the same alert threshold value for $RH_{ref} = 40\%$ and $80\%$ could yield the alert to be delayed by more than 4 hours.

5    To reduce the sensitivity of the alert to initial conditions, we propose to use the temporal gradient of $f_{\beta_{att}}$, $\partial f_{\beta_{att}}/\partial t$, as a proxy to monitor the hygroscopic growth dynamics. Figure 4 shows that the curves representing different initial conditions are much closer to each other for $\partial f_{\beta_{att}}/\partial t$ than for $f_{\beta_{att}}$. Hence the time at which a threshold is reached is less sensitive to the initial $RH_{ref}$ value if $\partial f_{\beta_{att}}/\partial t$ is used. In PARAFOG, $\partial f_{\beta_{att}}/\partial t$ is computed as the slope of a $k$-point linear fit following this equation:

$$RG_k(t_m, r_j) = \frac{\sum_{l=0}^{k-1}(t_{m-l}-\overline{t(t_m)}) \cdot (f_{\beta_{att}}(t_{m-l}, r_j, RH(t_m, r_i)) - \overline{f_{\beta_{att}}(t_m, r_j, RH(t_m, r_i))})}{\sum_{l=0}^{k}(t_{m-l}-\overline{t(t_m)})^2}$$
Eq. (18)

where: $\overline{t(t_m)} = \frac{1}{k}\sum_{l=0}^{k-1} t_{m-l}$

and: $\overline{f_{\beta_{att}}(t_m, r_j, RH(t_m, r_i))} = \frac{1}{k}\sum_{l=0}^{k-1} f_{\beta_{att}}(t_{m-l}, r_j, RH(t_m, r_i))$

The altitude where $RG_k(t_m, r_j)$ is maximum, $H_{max}(t_m)$, is used to monitor the level where the hygroscopic growth of

15    aerosol particles is fastest.

***Step 3 (Alert levels).*** When PARAFOG is turned "ON", $RG_k(t_m, r_j)$ is computed at each time step and each altitude. It is then compared to different threshold values to derive pre-fog alerts of different levels. These alerts are intended to provide warning prior to possible formation of radiation fog. The fog alert levels are defined as follows:

20    -    **None:** Even though PARAFOG is "ON", the ALC attenuated backscatter is not changing significantly compared to the reference conditions.

-    **Low:** the ALC attenuated backscatter is changing compared to the reference conditions. The atmospheric conditions support the development of radiation fog. The altitude $H_{max}(t_m)$ can change significantly from one time step to the next.



- **Moderate**: the ALC attenuated backscatter is changing compared to the reference conditions. The atmospheric conditions support the occurrence of radiation fog. The altitude $H_{max}(t_m)$ is found at a height that varies slowly from one time step to the next.

- **Severe**: the ALC attenuated backscatter is changing rapidly compared to the reference conditions (one order of magnitude more rapidly than for a low-level alert). Formation of droplets and subsequent fog is imminent.

- **Fog**: fog is detected by the ALC at an altitude ranging from 0 to 400 m agl.

The thresholds used to define low-, moderate- and severe-level alerts based on $RG_k(t_m, r_j)$ are shown in Table 2. They have been derived empirically using 10 cases from the SIRTA dataset. $RG_{60}(t_m, r_j)$ is derived using Eq. (18) as a slope based on 60 min of measurements, updated at each time step. 60 min allows significant backscatter ratio gradients to be detected while eliminating rapid fluctuations. This is consistent with physical time constants observed in radiative cooling condition. Using the threshold shown in Table 2 for low-level alerts, Fig. 4a reveals that a low-level alert would appear 30-60 min earlier for a reference relative humidity of 40% compared to 80%. A severe-level alert based on a 40% reference relative humidity is likely to appear about 10 min earlier than for an 80% reference condition.

The threshold used to define fog-level alert corresponds to an attenuated backscatter value typically observed as the near-surface horizontal visibility drops below 1 km. A time series of simultaneous and collocated CL31 attenuated backscatter and diffusiometer visibility at 20 m agl is shown in Fig. 5. This example (and many others not shown) show that when the visibility (measured at 550 nm) drops below 1 km, the 20-m attenuated backscatter (measured at 905 nm) exceeds $2 \times 10^{-4}$ m$^{-1}$ sr$^{-1}$. To compare, visibility-to-extinction parametrizations (e.g. Nebuloni 2005) yield an extinction of about $3 \times 10^{-3}$ m$^{-1}$ at 905 nm for a 550-nm visibility of 1 km (neglecting absorption by water vapor molecules). Because the extinction is strong in these conditions (scattering by droplets and absorption by water vapor), it is expected that the attenuated backscatter at higher altitudes (in the fog) will decrease quickly with increasing altitude. Hence this threshold will only be valid in the first few ALC measurement gates inside the fog.

***Step 4 (End).*** Several conditions can lead to the termination of a radiation fog alert event. PARAFOG considers the end of the fog if one of the following events occur: (1) occurrence of a cloud cover fraction greater than 50% in the range 400 m to





top of ALC range; (2) dissipation of cloud or fog layers in the 0-to-400 m range and (3) near-surface relative humidity

remaining below 85% for more than 10 min.

### 4.2. Case studies of radiation fog detected by PARAFOG

We test the performance of the PARAFOG algorithm on two situations prior to deep radiation fog formation at SIRTA and

UCCLE, as well as one situation prior to a shallow patchy radiation fog at SIRTA, and a quasi-fog at SIRTA. Figure 6 shows

the time series of measurements and pre-fog alert levels for the 15-16 November 2011 at SIRTA. Figure 6a shows the

calibrated range-corrected attenuated backscatter power provided by the CL31 and the near-surface horizontal visibility.

Reference $\beta_{att}$ are on the order of 1 x $10^{-6}$ m$^{-1}$ sr$^{-1}$ and $RH_{ref} = 77\%$ for this case. Figure 6b shows the pre-fog alert levels,

$H_{max}$, and the status of PARAFOG (ON/OFF). Eight hours prior to fog occurrence time, PARAFOG was already ON,

showing that the near-surface relative humidity exceeded 85% at 17:00UTC, while the horizontal visibility exceeded 10 km.

$H_{max}$ is found at an altitude slowly evolving between 50 to 200 m agl between 17:00 and 21:30 UTC. Then between 21:30

and 00:30 UTC, $H_{max}$ decreases slowly towards 100m agl. The first alert occurs at 00:20 UTC at 90 m agl, about 1 hour

prior to fog occurrence time. The alert-level evolves quickly from low to moderate to severe in less than 20 min. A fog layer

forms at 90 m agl at about 00:50 UTC just 20 min prior to fog occurrence at the ground. So once fog droplets appear at 90 m

agl, the fog layer starts to subside at a rate of about 300 m/h.

Figure 7 is the same as Fig. 6 for the case of 24 October 2012 at UCCLE. Reference $\beta_{att}$ are on the order of 1 x $10^{-6}$ m$^{-1}$ sr$^{-1}$

and $RH_{ref} = 60\%$ for this case. The PARAFOG algorithm status becomes "ON" 3 hours prior to fog occurrence time. Low-

level alerts appear soon after the algorithm turns "ON" with $H_{max}$ near 150 m agl. Consistent pre-fog alerts near 200m agl

occur at 21:45 UTC, 90 min prior to fog occurrence time. After 22:00 UTC, $H_{max}$ start to decrease quasi-monotonously at a

rate of about 100 m/h.

Figure 8 is the same as Fig. 6 for the case of 19-20 November 2011 at SIRTA. This is a shallow radiation fog situation with

patchy fog and highly variable horizontal visibility fluctuating between 200m and 2 km during ten hours. The PARAFOG

algorithm status is "ON" during the entire period. Low and moderate-level alerts occur at 10 m agl from 21:00 to 00:45

UTC. Moderate and severe-level alerts occur at 10 m agl from 00:45 to 06:00 UTC. $H_{max}$ values range predominantly 0-50

m agl, but values exceeding 300 m agl are also found nearly every hour.





Figure 9 is the same as Fig. 6 for the case of 09-10 December 2011 at SIRTA. High near-surface relative humidity, clear skies, and radiative cooling characterize this situation but fog formation does not occur. The PARAFOG algorithm status changes between "ON" and "OFF" several times during the ten-hour period. $H_{max}$ values are found anywhere between 0 and 400 m agl, showing that there is not a preferred altitude where aerosol activation is occurring. No pre-fog alert occur during

the ten-hour period.

## 5.   Performance of PARAFOG in providing pre-fog alerts

This Section presents the performance of the PARAFOG algorithm in terms of alert occurrences, durations, and altitudes. Results are provided for the low-, moderate-, severe-, and fog-level alerts based on about 45 fog cases observed near Paris

and Brussels presented in Table 3.

### 5.1.   Reference attenuated backscatter profile

The hygroscopic growth function, $f_{\beta_{att}}(RH)$, requires identifying a reference condition, from which the hygroscopic growth will be determined. If RH<40%, no hygroscopic growth may be considered (i.e., typically called dry conditions). If

RH<75%, the hygroscopic growth is already initiated although the size increase is considerably smaller than the increase from 75% to saturation. The algorithm described in Sect. 4.1 allows us to determine the reference attenuated backscatter profile $\beta_{att}(r_j, RH_{ref}(t_{ref}, r_i))$ for each pre-fog period. Figure 10 shows the distribution of reference $\beta_{att}$ profiles between 0 and 400 m agl obtained in pre-fog conditions at the time of $RH_{ref}$ (median, mean and quartiles) at SIRTA (Fig. 10a,b) and UCCLE (Fig. 10c,d). Figure 10a,c shows profiles for which the minimum $RH$ near-surface was greater than 75%, while Fig.

10b,d corresponds to minimum $RH$ less than 75%. For the driest reference conditions (near-surface RH < 75%), the average attenuated backscatter value (905 nm) is about 1 x $10^{-06}$ sr$^{-1}$ m$^{-1}$ at both SIRTA and UCCLE. This corresponds to near-surface horizontal visibility greater than 20 km. These profiles appear to be well mixed between 0 and 250 m agl, with very little attenuation (T$^2$ > 0.95, hence the extinction is less than 0.85 x $10^{-04}$ m$^{-1}$). The decrease in $\beta_{att}$ above 300 m agl is most likely due to a decrease in aerosol concentration above that layer. Note that at SIRTA a peak attenuated backscatter value is

observed at 50 m agl. This is a known opto-electronic measurement artifact that is discussed in Kotthaus et al (2016). At





UCCLE, $\beta_{att}$ measurements of the CL51 below 50 m are not used because they could not reliability track the fog formation. For the moistest reference conditions (near-surface RH > 75%), the average surface attenuated backscatter can exceed 2 x $10^{-06}$ sr$^{-1}$ m$^{-1}$, corresponding to horizontal visibility of about 7-8 km. The average profiles reveal a strong attenuation between the surface and the top of the moist layer near 350 m agl, with a mean extinction near 2 x $10^{-03}$ m$^{-1}$. This extinction is
partially due to scattering by moist aerosols and absorption by water vapor bands at 905 nm (Wiegner and Gasteiger, 2015).

### 5.2. Alert occurrences

The time at which fog occurrence starts at ground level (< 10 m agl) is called *fog occurrence time*. Prior to this time various alert levels may occur at altitudes ranging 0 to 400m agl. Figure 11 shows, for each altitude agl and each time before *fog*

*occurrence time*, the frequency of occurrence of each alert level including no-alert or low-, moderate-, severe-, and fog-level alerts. For each height / time interval, the sum of occurrences of all alert levels = 100% (incl. no alert). Figure 11a and b represent the alert occurrences based on SIRTA and UCCLE data respectively.

We find that moderate-level alerts occur either near the ground (<10 m agl) or aloft (50-100 m agl), starting more than 120 min prior to *fog occurrence time*. 30-10 min prior to *fog occurrence time*, the frequency of occurrence of moderate-level

alerts increases, with higher (in terms of altitude) alerts occurring earlier.

Severe-level alerts occur generally less than 50 min (100 min) prior to *fog occurrence time* at SIRTA (UCCLE). SIRTA data show that 15 min prior to fog occurrence time the severe-level alert occurrence is most frequent at 70 m agl. The severe-level alert then occurs progressively at lower altitude as the time prior to *fog occurrence time* decreases. The fog forms at or very close to the surface (< 20 m agl) after this severe-level alert scenario. A similar result is found based on UCCLE data with

severe-level alert occurrences exceeding 50% about 40 min prior to *fog occurrence time* at altitudes ranging 50-100 m agl. Severe-level alerts at SIRTA occur also at altitudes above fog-level alerts (typically > 70m). These occurrences should likely be fog-level alerts, but at these altitudes the ALC signal does not exceed the fog-level threshold if there is a fog layer below due to strong attenuation.

The fog-level alert is a scenario where the liquid water phase occurs aloft prior to fog occurrence at the ground. This

occurrence reveals a time/height relationship showing a progressive increase in the subsidence rate of the fog layer in the few minutes before it reaches the ground.




### 5.3. Alert durations

Alert durations are computed for moderate-, severe-, and fog-level alerts. The total alert duration is the time interval between the beginning of a moderate-level alert and the *fog occurrence time*. During this time interval, moderate-, severe- and fog-level alerts are present. The sum of the moderate, severe and fog alert durations corresponds to the total alert duration. The moderate-level alert duration is the time interval during which a moderate-level alert is present. A severe-level alert duration is the time interval during which a severe-level alert is present. A Fog-level alert duration is the time interval during which a fog-level alert is present. Note that the fog-level alert stops when the *fog occurrence time* is reached.

Figure 12 shows that moderate-level-alert durations last 5-20 min in general at SIRTA and UCCLE. They represent about 5 to 30% of the total alert duration. But at SIRTA on some occasions, the moderate-level-alert duration can represent more than 50% of the total alert duration. Severe-level-alert durations last up to 70 min at UCCLE, while they are shorter at SIRTA. They can represent up to 80% of the total alert duration. Fog-level-alert durations last up to 30 (50) min at SIRTA (UCCLE), representing up to 90% of total alert durations. This confirms that alerts prior to fog occurrence at the ground can follow different scenarios, some of which include a long fog-level alert aloft prior to fog occurrence at the ground.

Figure 13 shows the effect of $RH$ that was present at the time when the reference $\beta_{att}$ is chosen. While this $RH_{ref}$ is measured near the ground, we concluded from Fig. 10 that it is a good proxy for $RH$ throughout the shallow nighttime mixing layer. For both sites, we find a strong negative correlation between the total alert duration and $RH_{ref}$. This is consistent with the behavior shown in Fig. 4c, illustrating that alerts occur later when reference conditions are more humid. At SIRTA (UCCLE) alert durations are reduced by 5 (10) minutes when $RH_{ref}$ increases by 10%. Figure 13 possibly reveals two populations. The first population is characterized by shorter alert durations (typically less than 50 min) for which alert durations are reduced by 5 min when $RH_{ref}$ increases by 10%. The second population is characterized by longer alert durations (typically greater than 50 min) for which alert durations are reduced by 10 min when $RH_{ref}$ increases by 10%. Pre-fog alerts at SIRTA are found predominantly in the first population, while at UCCLE alerts are found in both populations.

### 5.4. Alert altitudes





Next we define the altitude, $H_{max}$, where the maximum ratio gradient occurs at a given time. This parameter traces at which altitude aerosol activation is strongest, identifying the altitude where the highest supersaturation occurs. $H_{max}$ is determined only when an alert exists. In case of low-level alerts, $H_{max}$ altitudes are found near the surface and aloft, with rapid changes between different altitudes from one time step to the next. In case of moderate-, severe- and fog-level alerts, $H_{max}$ altitudes

behave more consistently with time. Hence $H_{max}$ is only determined for moderate-, severe- and fog-level alerts. Figure 14 shows a frequency distribution function of $H_{max}$ derived from SIRTA and UCCLE data. $H_{max}$ ranges predominantly between 0 and 300m, with a median (mean) value of 75 (91) m at SIRTA and 110 (124) m at UCCLE. Lowest possible $H_{max}$ at UCCLE is 50 m because CL51 data are not exploited below this altitude. Although measurements start at 50 m, a possible explanation for the higher $H_{max}$ values at UCCLE compared to SIRTA is that UCCLE is an urban site, located only 5 km

from the city center, while SIRTA is suburban located 20 km South-West of the city center. The urban site is subject to higher turbulence near the surface due to surface heterogeneities. The 3-D structure of the city canopy generates small-scale turbulence that tends to mix the air in the shallow surface layer. It slows down the cooling of this layer and makes it more difficult to reach supersaturated conditions at the surface. Upward vertical motions can lead to cooling of rising moist plumes, enabling supersaturated conditions to be reached a few tens or hundreds of meters above the ground.

Next we study the rate of change of $H_{max}$, computed based on 10-min time intervals. The frequency distribution function of the rate of change of $H_{max}$ is shown in Fig. 15, for moderate-, severe-, and fog-level alerts, based on SIRTA and UCCLE data. When the rate of change is negative (positive), $H_{max}$ decreases (increases) with time. Figure 15 shows that temporal rate of change (m/hr) of $H_{max}$ ranges predominantly between 0 and -300 m/hr. Positive values are also observed. During moderate-level alerts, the rate of change of $H_{max}$ is found to range predominantly between +/- 50 m/hr, with a median (mean)

value of 0 (-25) m/hr at SIRTA and -60 (-67) m/hr at UCCLE. This reveals that moderate-level alert heights decrease slowly as time approaches the *fog occurrence time*. During severe-level alerts the median (mean) rate of change of $H_{max}$ is -60 (-70) m/hr at SIRTA and -60 (-49) m/hr at UCCLE. So when a severe-level alert is reached, the height, at which this alert occurs, decreases significantly as time approaches the *fog occurrence time*. When fog-level alerts occur, the median (mean) rate of change of $H_{max}$ is -125 (-114) m/hr at SIRTA and -60 (-102) m/hr at UCCLE, revealing the most rapid rate of decrease of the

alert height prior to fog occurrence at the surface.



## 6. Conclusions

We demonstrate that total attenuated backscatter in the near-IR can be used as a proxy to track aerosol backscatter coefficient hygroscopic growth. We show that hygroscopic growth functions of aerosol backscatter can be derived using a simple

experimental setup including an ALC and a hygrometer on a 30-m mast. The hygroscopic growth function is defined as the ratio of attenuated backscatter to a reference attenuated backscatter measured in dry conditions. These functions are similar to those found in the literature for other aerosol optical properties. A reliable reference attenuated backscatter profile must be found to track hygroscopic growth of aerosol backscatter. The reference attenuated backscatter profile varies significantly with relative humidity conditions. It has been found that RH < 75% produce more reliable reference attenuated backscatter

profiles.

In pre radiation fog conditions, one usually finds that RH increases with time resulting in an increase of attenuated backscatter. Hence the temporal gradient of the attenuated backscatter ratio can be derived. We find that the temporal gradient of attenuated backscatter ratio is less sensitive to initial conditions than the attenuated backscatter ratio itself. Temporal gradients of attenuated backscatter ratio reach threshold values showing that the aerosol activation process is

engaged. In pre radiation fog conditions, we find that these thresholds are reached up to 1-3 hours prior to the time of fog occurrence at altitudes ranging from 0 to 250 m agl.

$H_{max}$, the altitude at which the maximum gradient of attenuated backscatter ratio, is another useful parameter. It reveals at which altitude cooling processes most efficiently lead to aerosol activation. It often reveals that $H_{max}$ tends to decrease as time reaches the fog occurrence time. We find that $H_{max}$ is quite variable with time in situations prior to very shallow and

patchy radiation fog. Conversely, $H_{max}$ is rather stable with time prior to deep radiation fog.

Based on measurements carried out at two stations in urban and semi-urban environments, we find that pre-fog alert occurrences and durations depend on the cooling processes leading to supersaturated conditions, and on the reference conditions that can be found. Hence alert thresholds should be adjusted for each site. Alert durations could be optimized by adapting alert thresholds to relative humidity values found at the reference time. Based on 45 cases studied and given alert

thresholds, alert occurrence reach 100% thirty (fifty) min prior to fog formation at SIRTA (UCCLE).





The PARAFOG algorithm is currently deployed at the Charles-de-Gaulle airport in Paris, using CL31 measurements, providing airport forecasters one additional source of information to help them anticipate occurrences of low-visibility runway conditions.

Further analyses should be carried out to test the performance of PARAFOG at other locations. Alert threshold values should

5  be adapted to reference relative humidity, and possibly to aerosol types using for example PM2.5 measurements.

**Data availability**

CL31 data, surface meteorological parameters and visibility measurements at SIRTA can be access from the SIRTA public data repository that is accessible online at http://www.sirta.fr. The data policy and a data download are available from the website. CL51 data and surface meteorological parameters are available on request (laffineur.quentin@meteo.be).

**Author contribution**

M. Haeffelin and Q. Laffineur designed the experiment and developed the PARAFOG algorithm. J-A Bravo-Aranda carried out the simulations presented in Section 3. J-A. Casquero-Vera tested the algorithm on a subset of the SIRTA CL31 dataset. M-A. Drouin coded the PARAFOG algorithm in Python and adapted it to run automatically on large datasets. J-C. Dupont provided access to SIRTA data and H. DeBaker to UCCLE data. M. Haeffelin prepared the manuscript with contributions

from all co-authors.

**Acknowledgements**

This study was conducted in the framework of the TOPROF (ES1303; http://www.toprof.imaa.cnr.it/) European COST action. The authors would like to acknowledge the EU COST office for its support for scientific meetings and missions. ES1303 COST. The authors would like to acknowledge Météo-France for providing the CL31 ceilometer deployed at the

SIRTA observatory and the Solar-Terrestrial Centre of Excellence (STCE), a research collaboration established by the Belgian Federal Government through the action plan for reinforcement of the federal scientific institute that funded the CL51 ceilometer deployed in UCCLE.




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



**TABLES**

|  | Laser Source | Output Laser Energy | Pulse Rate | Altitude Range | Vertical Resolution | Full Optical Overlap |
|---|---|---|---|---|---|---|
| Vaisala CL31 | InGasAs 910 nm | 1.2 µJ | 10 kHz | 0m - 7.5km | 15 m | < 50 m |
| Vaisala CL51 | InGasAs 910 nm | 3 mJ | 6.5 kHz | 0m – 15km | 10 m | < 500 m |

**Table 1. Main characteristics of the CL31 (SIRTA) and CL51 (UCCLE) used in this study.**



| | $RG_{60}(t_m, r_j)$ | $Pr^2(t_m, r_j)$ | $RH(t_m, r_i)$ |
| --- | --- | --- | --- |
| | $(s^{-1})$ | $(m^{-1}\ sr^{-1})$ | (%) |
| Low | $4 \times 10^{-4}$ | | 90 |
| Moderate | $1 \times 10^{-3}$ | | 95 |
| Severe | $4 \times 10^{-3}$ | | 95 |
| Fog | | $2 \times 10^{-4}$ | |

5    **Table 2. Thresholds used to define low-, moderate- and severe-level alerts based on ALC attenuated backscatter ratio gradients.**





| (a) SIRTA Dev. Fog | | | | |
|---|---|---|---|---|
| Date (Day/Month/Year) | Fog occurrence time (hh:min) | Alert duration (min) | $RH_{ref}$ (%) | Initial value of $H_{max}$ (m) |
| 13-02-2011 | 04:00 | 20 | 79 | 75 |
| 10-11-2011 | 18:00 | 16 | 73 | 90 |
| 15-11-2011 | 03:00 | 32 | 61 | 135 |
| 16-11-2011 | 01:20 | 24 | 77 | 90 |
| 25-11-2011 | 21:40 | 21 | 98 | 75 |
| 28-11-2011 | 06:30 | 23 | 85 | 120 |
| 19-12-2011 | 05:10 | 46 | 63 | 45 |
| 23-10-2012 | 03:40 | 107 | 59 | 300 |
| 24-10-2012 | 03:40 | 68 | 73 | 120 |
| 04-11-2012 | 01:10 | 23 | 62 | 150 |
| 09-11-2012 | 06:40 | 11 | 74 | 45 |
| 14-11-2012 | 09:30 | 67 | 99 | 195 |
| 20-11-2012 | 03:00 | 4 | 99 | 60 |
| 22-11-2012 | 03:40 | 16 | 79 | 60 |
| 30-11-2012 | 19:00 | 4 | 98 | 90 |
| 10-03-2013 | 05:00 | 24 | 55 | 105 |
| 24-10-2013 | 03:30 | 36 | 66 | 120 |
| 10-01-2014 | 03:20 | 10 | 71 | 75 |
| 11-01-2014 | 09:00 | 3 | 97 | 30 |
| 31-01-2014 | 00:40 | 8 | 95 | 60 |
| 22-02-2014 | 23:10 | 14 | 71 | 135 |
| 14-03-2014 | 23:20 | 47 | 39 | 135 |
| 28-10-2014 | 00:30 | 23 | 68 | 60 |
| 27-11-2014 | 00:50 | 2 | 92 | 0 |
| 30-12-2014 | 07:50 | 19 | 73 | 30 |
| 03-01-2015 | 01:30 | 7 | 74 | 75 |
| 07-01-2015 | 00:50 | 16 | 71 | 75 |
| 12-02-2015 | 05:50 | 15 | 62 | 30 |
| 18-02-2015 | 05:40 | 36 | 74 | 75 |



| (b) Uccle Dev. Fog | | | | |
|---|---|---|---|---|
| Date (Day/Month/Year) | Fog occurrence time (hh:min) | Alert duration (min) | $RH_{ref}$ (%) | Initial value of $H_{max}$ (m) |
| 27-02-2012 | 06:10 | 33 | 78 | 120 |
| 04-03-2012 | 02:00 | 76 | 67 | 200 |
| 04-04-2012 | 06:20 | 75 | 39 | 160 |
| 04-05-2012 | 05:30 | 63 | 76 | 110 |
| 20-05-2012 | 22:20 | 168 | 57 | 170 |
| 24-10-2012 | 23:30 | 91 | 60 | 190 |
| 04-11-2012 | 05:40 | 21 | 67 | 140 |
| 18-11-2012 | 19:30 | 8 | 87 | 130 |
| 16-01-2013 | 21:30 | 42 | 56 | 160 |
| 23-01-2013 | 00:10 | 57 | 92 | 80 |
| 31-05-2013 | 00:20 | 48 | 79 | 180 |
| 23-09-2013 | 02:00 | 173 | 61 | 200 |
| 27-09-2013 | 02:50 | 20 | 53 | 110 |
| 24-10-2013 | 05:00 | 80 | 64 | 110 |

5   **Table 3. Fog events used to study the occurrence and characteristics of pre-fog alerts (a) at SIRTA, (b) at UCCLE.**





**Figures**

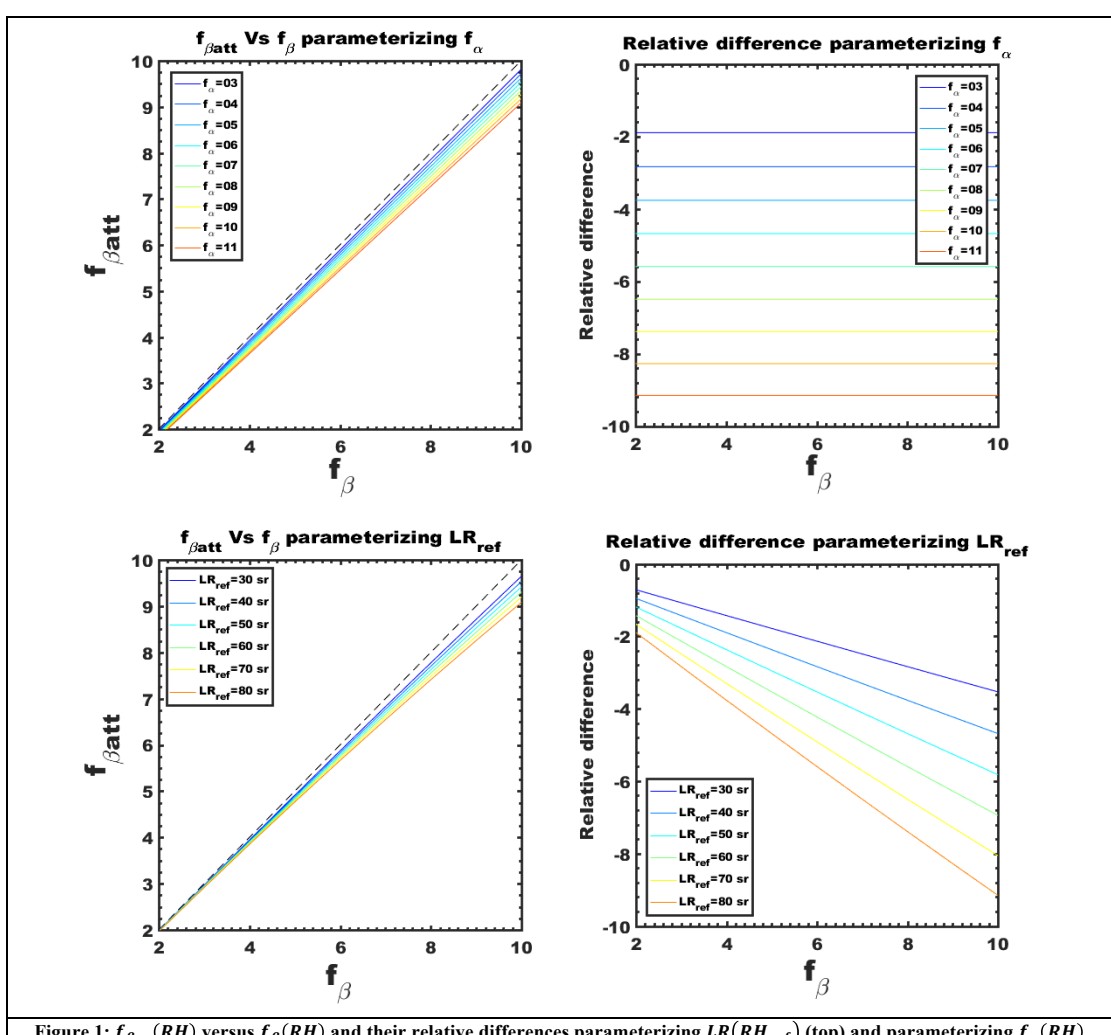

**Figure 1:** $f_{\beta_{att}}(RH)$ versus $f_\beta(RH)$ and their relative differences parameterizing $LR(RH_{ref})$ (top) and parameterizing $f_\alpha(RH)$ (bottom). Parameterization values are shown in the labels. 1:1 lines are shown as dashed blue line where relevant.





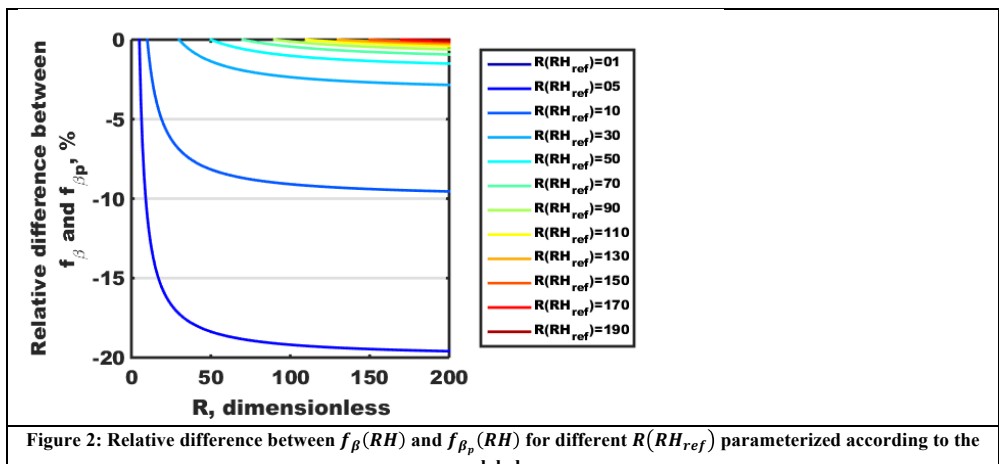

**Figure 2: Relative difference between $f_\beta(RH)$ and $f_{\beta_p}(RH)$ for different $R(RH_{ref})$ parameterized according to the label.**




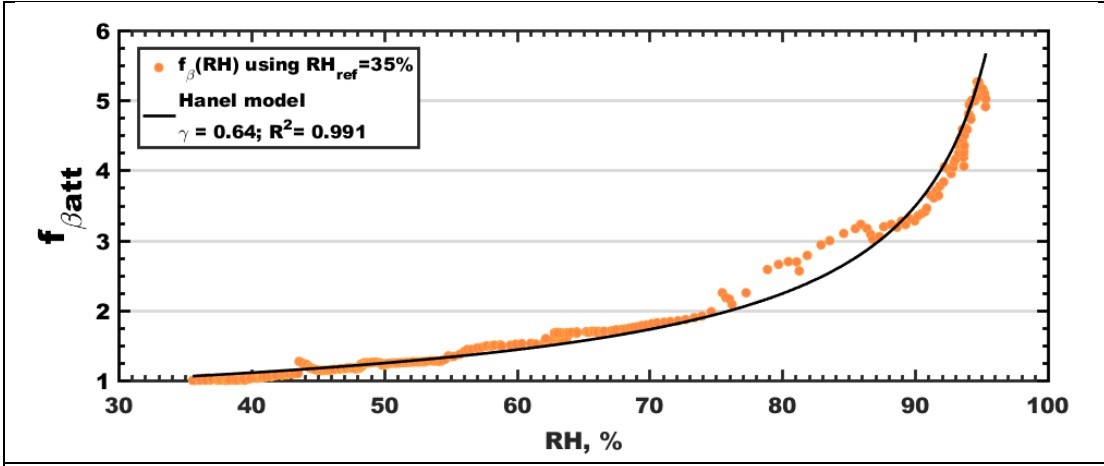

**Figure 3:** $f_{\beta_{att}}(RH)$ **as function of relative humidity (orange dots) measured at 30 m agl on 10 March 2013 at SIRTA and its Hänel function fit (black line).**



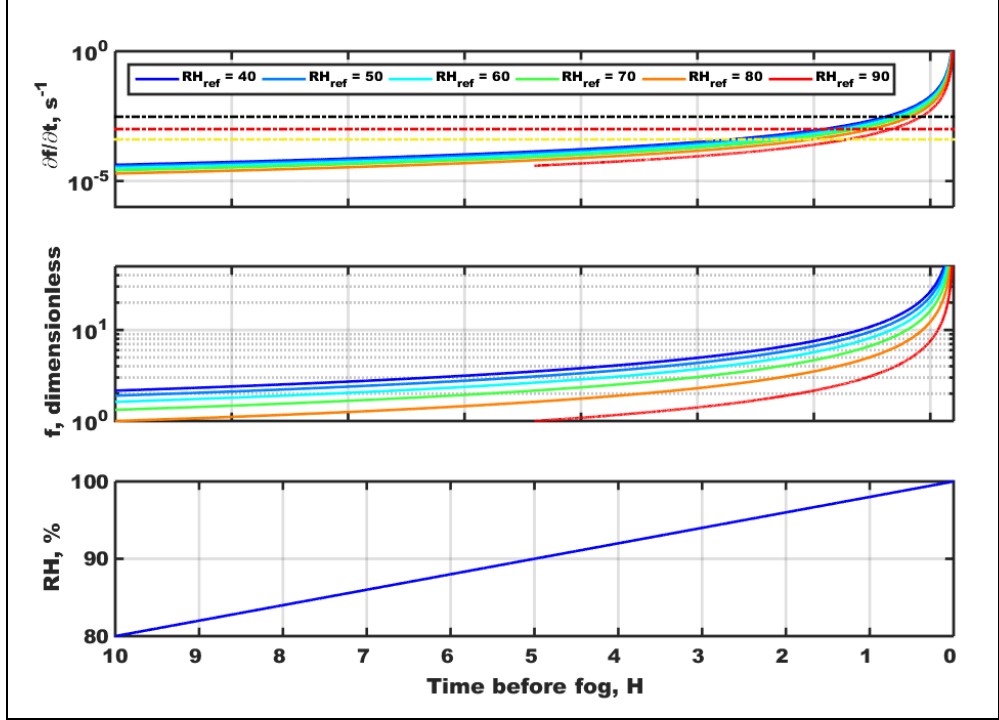

**Figure 4:** Temporal evolution of $\partial f(RH)/\partial t$ (top), $f(RH)$ (center), and $RH$ (bottom) based on Eq. 4 and Eq. 18 for different $RH_{ref}$ according to the label and $\gamma = 0.7$.





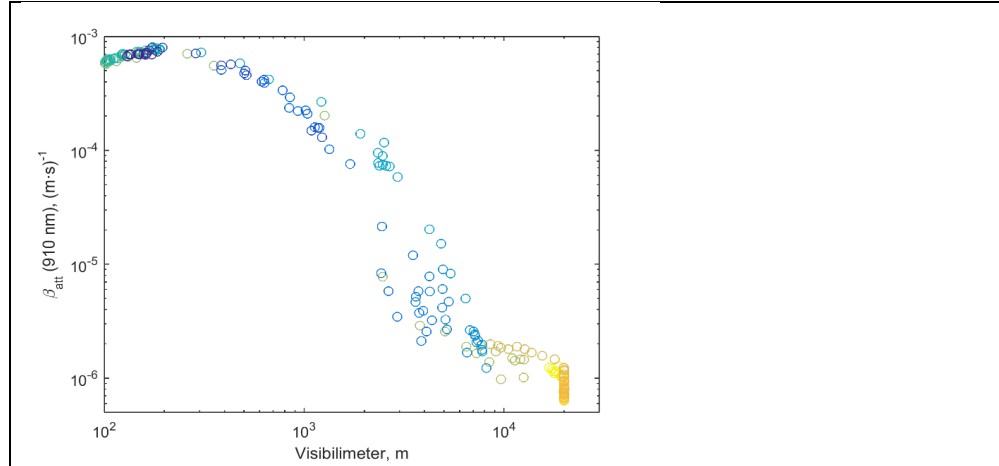

**Figure 5: time series of simultaneous and collocated CL31 $\beta_{att}$ (905 nm) and diffusiometer visibility (550 nm) measured at 20 m agl**





FIGURES SECTION 4: Fog and Quasi-fog cases

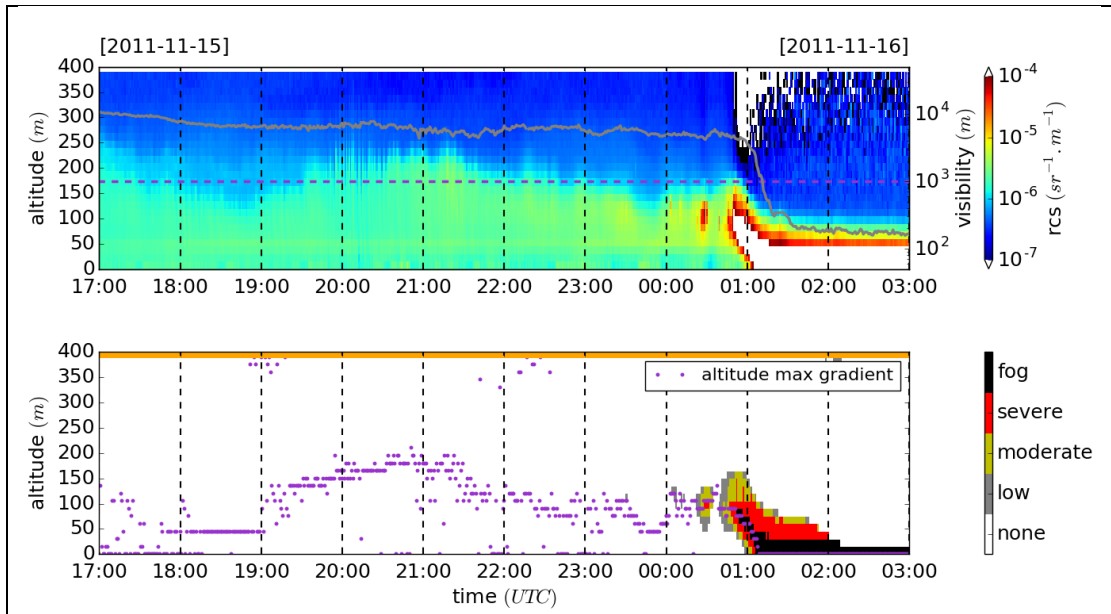

**Figure 6: time series presenting measurements and alert levels in pre-fog conditions on 15-16 November 2011 at SIRTA. (a) ALC attenuated backscatter and horizontal visibility. (b) Alert levels and altitude of $H_{max}$. Pre-fog conditions are clear. Aerosol activation occurs at 100 agl. Low-level alert 90 min before fog formation. Severe alert 30 min before fog formation.**





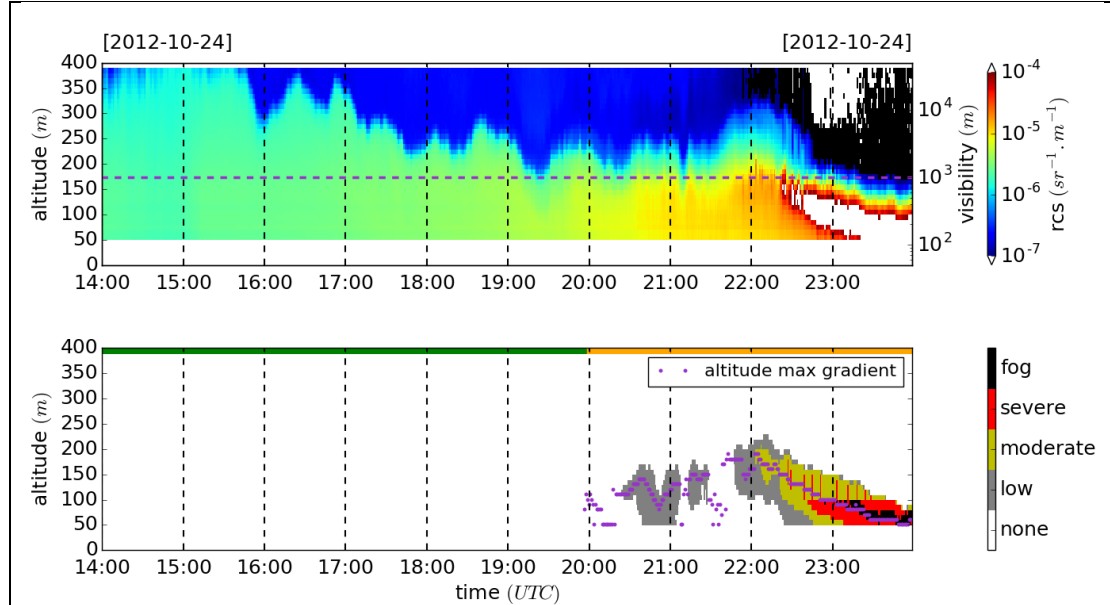

**Figure 7:** time series presenting measurements and alert levels in pre-fog conditions on 24 October 2012 at UCCLE. (a) ALC attenuated backscatter and horizontal visibility. (b) Alert levels and altitude of $H_{max}$. Pre-fog conditions are clear. Aerosol activation occurs at 100 agl. Low-level alert 90 min before fog formation. Severe alert 30 min before fog formation.





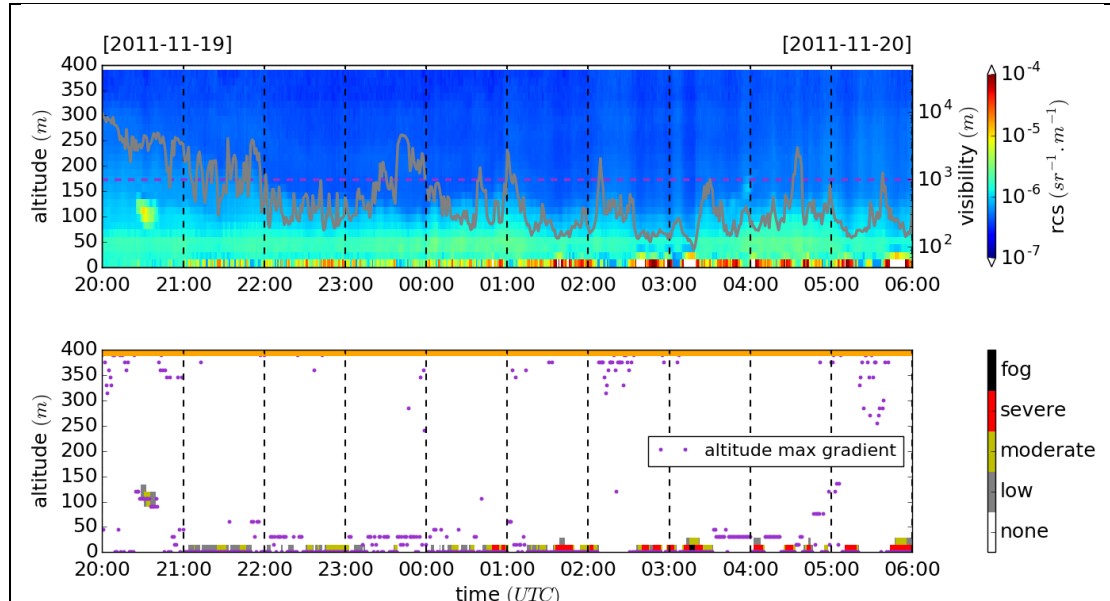

Figure 8: time series presenting measurements and alert levels in pre-fog conditions on 19-20 November 2011 at SIRTA. (a) ALC attenuated backscatter and horizontal visibility. (b) Alert levels and altitude of $H_{max}$. Pre-fog conditions are clear. Aerosol activation occurs at 0 m agl. Low-level alert 90 min before fog formation. Severe alert 30 min before fog formation.





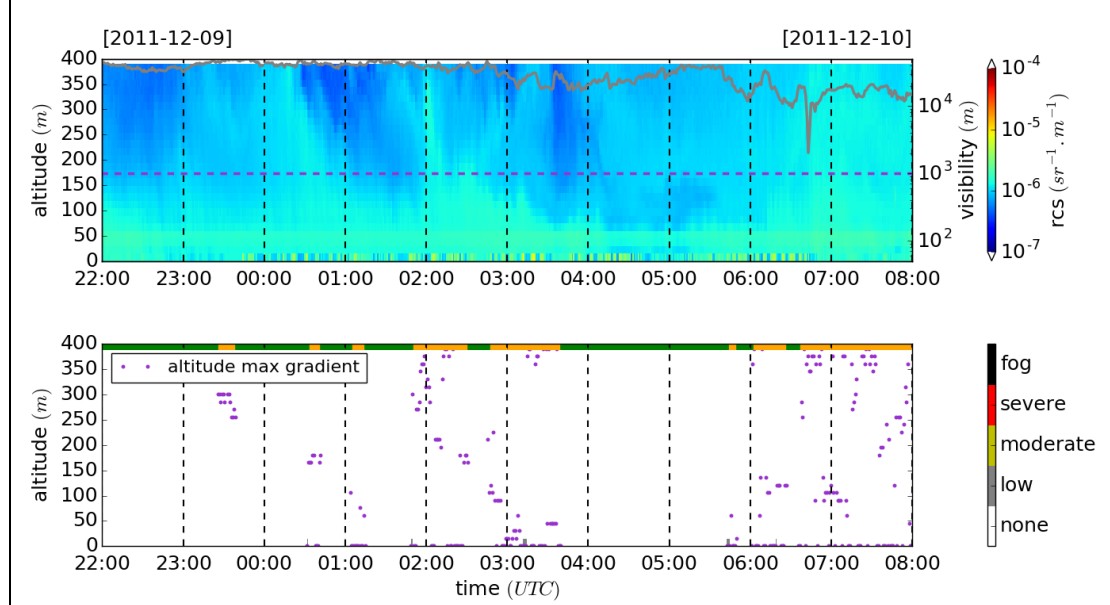

**Figure 9: time series presenting measurements and alert levels in pre-fog conditions on 09-10 December 2011 at SIRTA. (a) ALC attenuated backscatter and horizontal visibility. (b) Alert levels and altitude of $H_{max}$. Pre-fog conditions are clear.**





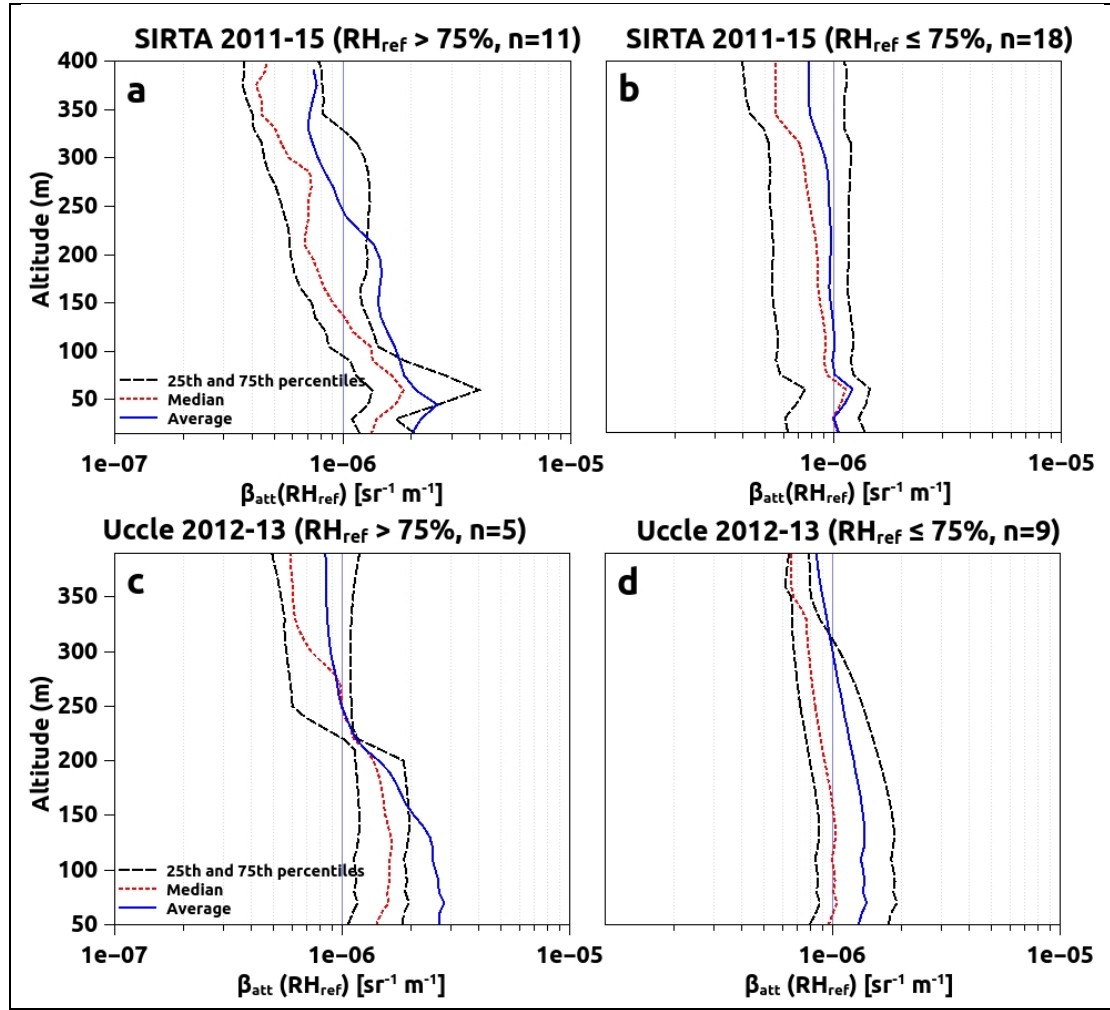

**Figure 10: Distribution of reference $\beta_{att}$ profiles between 0 and 400 m agl obtained in pre-fog conditions at the time of $RH_{ref}$ (median, mean and quartiles according to the label) at SIRTA (a,b) and UCCLE (c,d for which the minimum $RH$ near-surface was greater than 75% (a,c) and less than 75% (b,d).**



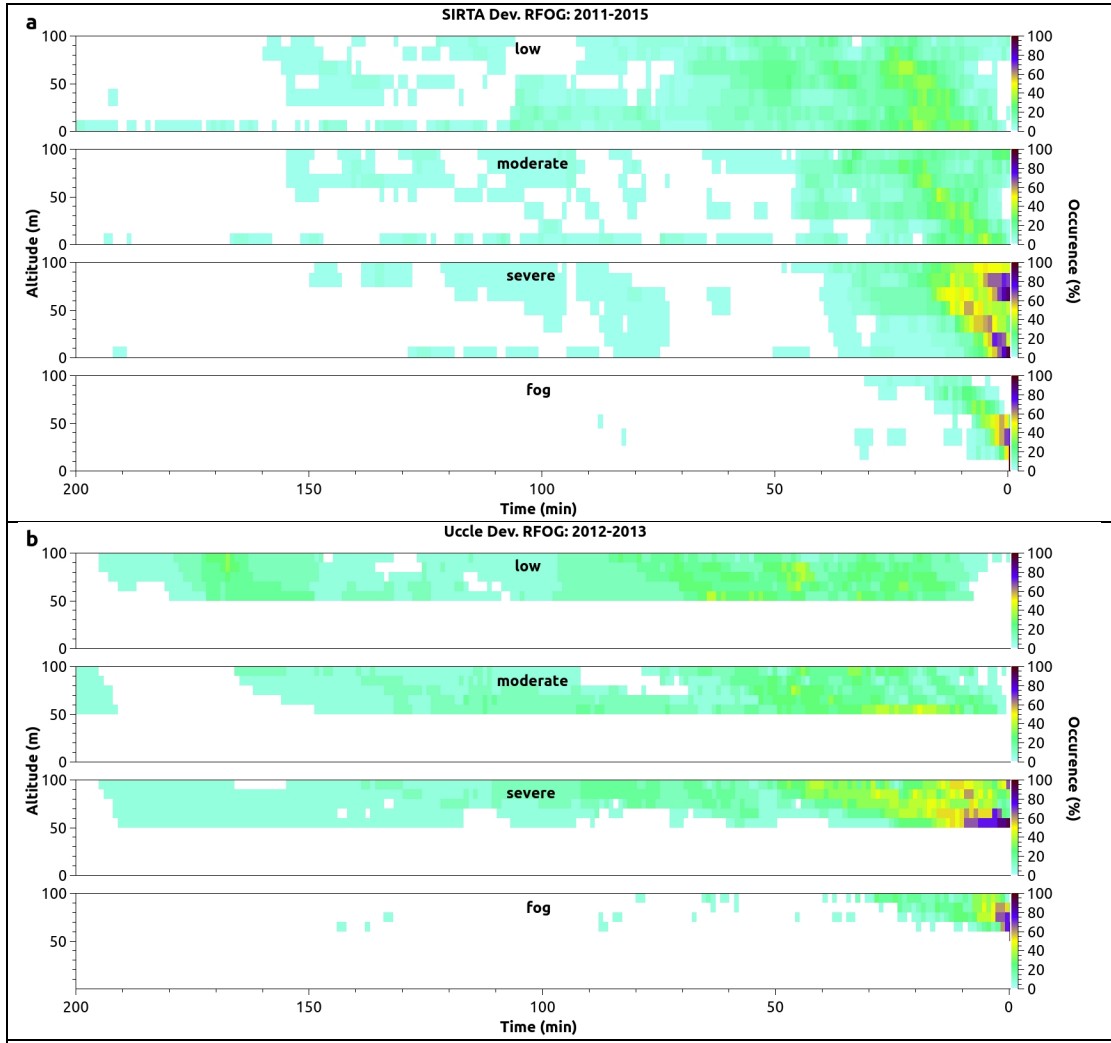

**Figure 11: Frequency of occurrence of each alert level (no-alert or low, moderate, severe, fog alerts) for each altitude agl and each time before fog occurrence time at SIRTA (a) and UCCLE (b). For each height / time interval, the sum of occurrence of all alert levels = 100% (incl. no alert).**





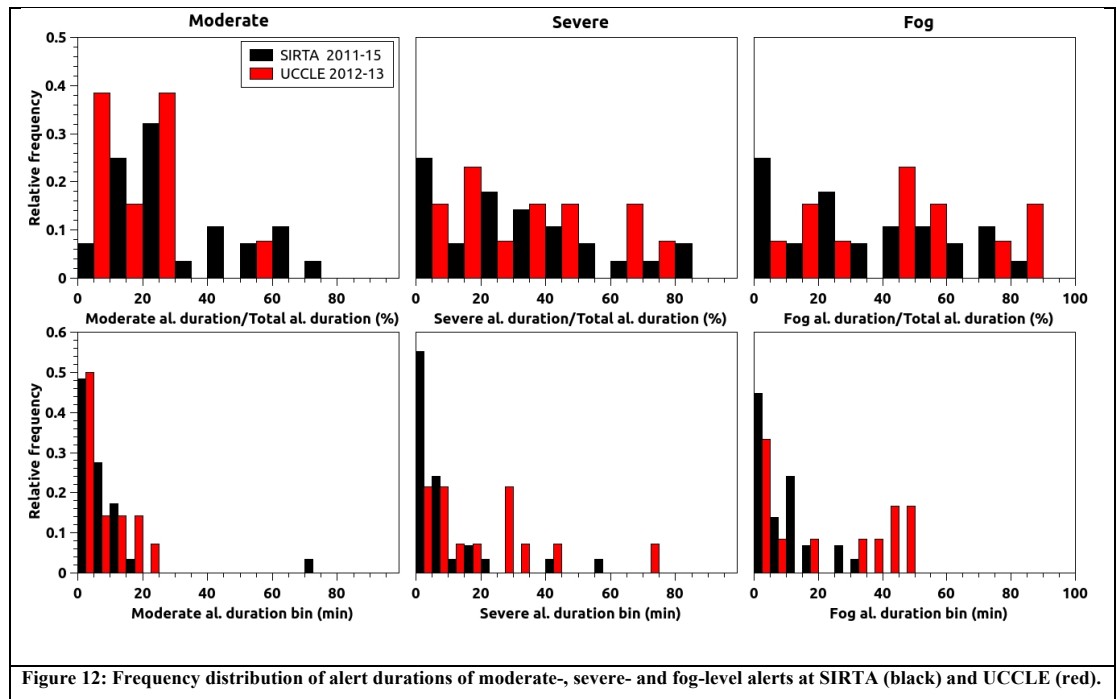

Figure 12: Frequency distribution of alert durations of moderate-, severe- and fog-level alerts at SIRTA (black) and UCCLE (red).



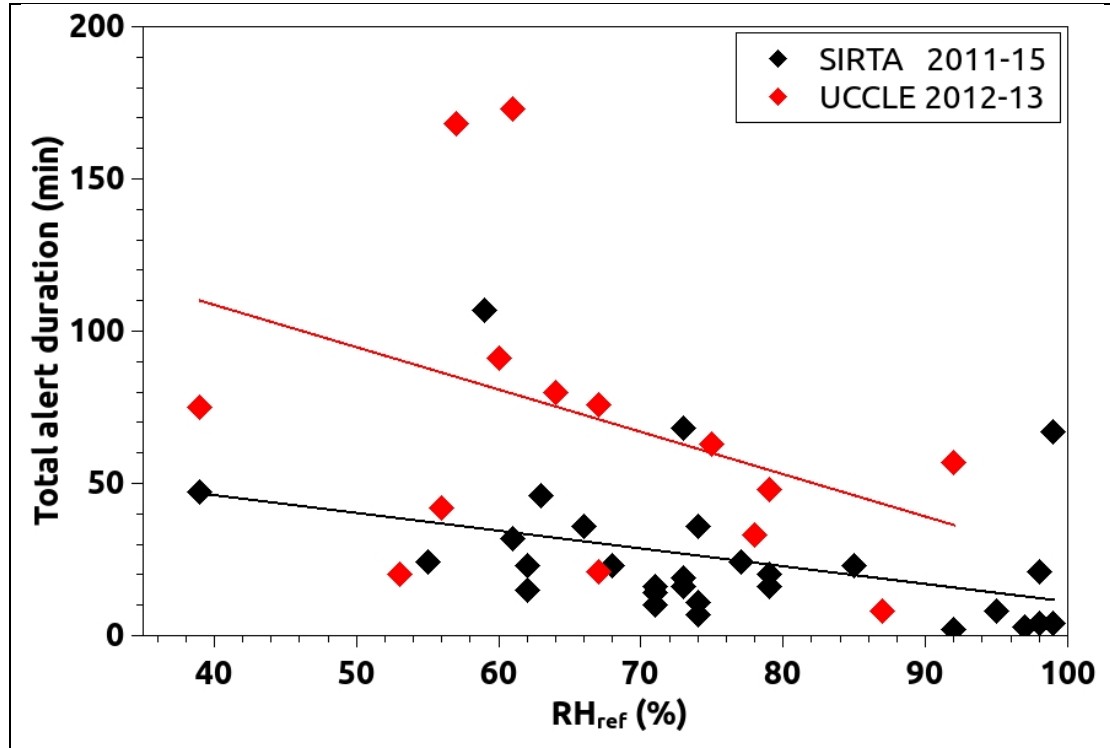

**Figure 13: Effect of *RH* at the time when the reference $\beta_{att}$ profile is derived on the total alert duration, at SIRTA (back) and UCCLE (red).**





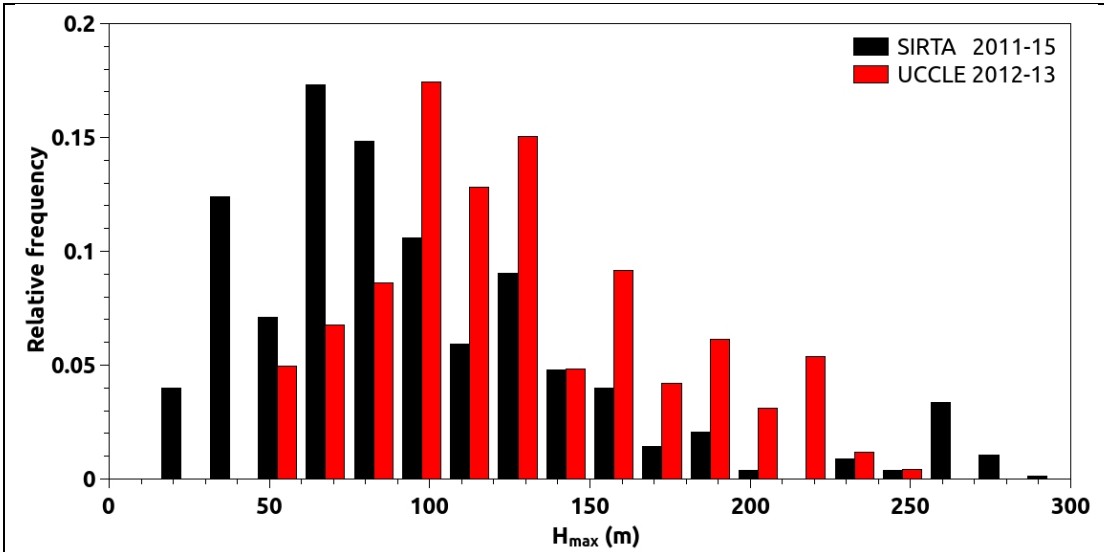

**Figure 14: Frequency distribution of the altitude where the maximum ratio gradient occurs at a given time. $H_{max}$ is determined only when moderate-, severe- or fog-level alerts exist.**







**Figure 15:** **Frequency distribution of rate of change of $H_{max}$ computed on a 10-min time interval. When the rate of change is negative (positive), $H_{max}$ decreases (increases) with time.**