# Peer review of "Radiation fog formation alerts using attenuated backscatter power from automatic Lidars and ceilometers"

_Atmospheric Measurement Techniques, 2016_

## Referee Comment (RC1) · Anonymous Referee #1 · 19 Jul 2016

Recommendation: minor revision

This paper presents a very interesting method for the nowcasting of radiation fog using measurements from Lidars and ceilometers, as well as some additional instrumentation. The findings are of great interest for the forecasting of radiation fog, especially in areas with availability of these instruments (e.g. airports, where the problems associated with fog are especially important). The paper is well structured, although other options for the structure of the paper are possible. A good introduction to the algorithm is presented, where the authors show clearly all the problems/solutions for the applicability of the method. Then, the authors present more specific analyses for four case studies, as well as several results for 45 cases of radiation fog at two locations. I

strongly recommend the publication of this paper in Atmospheric Measurements Techniques journal, since it provides a clear, good, interesting and potentially applicable method for the nowcasting of radiation fog. However, I have also some minor comments that could be addressed before the final publication.

GENERAL COMMENTS

- The Introduction section provides a good overview of the current problems in fog forecasting and the process of hygroscopic growth of fog condensation nuclei. There is a lack of references from Âą lines 4 to 12 of page 4. I recommend to include more references here.

- Section 2 introduces the sites, the differences between the instrumentation used at SIRTA and UCCLE. In this section, I think that section 2.3 (fog dataset description) is more a description of types of radiation fogs than a description of the cases used in the work, which is given in Table 3. Maybe the authors could re-consider the inclusion of this subsection in the final paper.

- Section 3 is the most theoretical part of the paper, where relationships of relative humidity, aerosol properties and measurements from ALCs are given. In this section, it is also explained technically how the hygroscopic growth of aerosols can be derived from attenuated backscatter profiles of ALCs and RH measurements from a hygrometer. Since I am not an expert on this, I cannot correct or make suggestions on the development of these relationships and on the experiments performed. However, I have found this section very interesting and clear for readers not familiar with these relationships.

- In Section 4, an algorithm to detect rapid changes in aerosol backscatter is presented, which is used to detect pre-fog periods.

- In step 1 of the algorithm (start), the authors could also suggest the use of previous works on detection of pre-fog periods to complement or improve this step, such those works presented by Menut et al. (2014) (DOI 10.1007/s10546-013-9875-1) or in

Román-Cascón et al. (2016) (DOI: 10.1002/qj.2708). The results used in these works could be a good complement to improve this step, although more instrumentation would be required. I think the authors could suggest this in this part.

- Briefly, all the steps and thresholds described and determined in the work seem to be quite studied and analysed by the authors, which demonstrate that the preparation of the work have been done in a proper way.

- In subsection 4.1, the authors test the algorithm over 3 situations at SIRTA and 1 at UCCLE.

- Figures 6, 7, 8, 9: I would recommend to include more details about the figures in the figure captions. For example: "Figure 6: Time series presenting measurements and alert levels in pre-fog conditions on 15-16 November 2011 at SIRTA. (a) ALC attenuated backscatter (colour contours) and horizontal visibility (grey line). (b) Alert levels (colours) and altitude of Hmax (purple points)."

- Figures 6, 7, 8, 9: Please, add height of measurement of horizontal visibility. Add also the meaning of horizontal green and orange lines in figure captions.

- Figure 7: Where is the line of visibility?

- Figure 8: "Low-level alert 90 min before fog formation. Severe alert 30 min before fog formation" (is this correct?)

- In the whole text, I think the term "low-level" (referring to alert level) can lead to confusion (alert in the low-level atmosphere). I recommend to use the term "minor" or another one to avoid this.

- Case studies of section 4.2 and section 5.4 (alert altitudes): It is shown how in many cases of radiation fogs studied, the activation of the droplets occurs at certain height at both sites (around 100 m (mean) and even higher at the urban site) and then it is observed the subsidence of Hmax. For me, it is not clear if this also means the subsidence of the "fog layer", as written in page 19 (lines 24 to 26) or only the activation. If the results refer to the subsidence of the fog layer, I wonder if this could be checked with different visibilimeters at different heights (maybe not available) or if this result could be an artefact from the algorithm or from the ALC measurements (maybe with more problems with layers below 70-100 m (not only at UCCLE)).These results are quite interesting, since normally (and based on other studies), radiation fog normally starts to be formed quite close to the ground and then it evolves into higher heights, except for "cloud-base-lowering (CBL) fog types", when a fog occurring certain day is the result of the lowering of "elevated" clouds from previous fog of the precedent day or even the result of the dissipation of previous fog (but with more avail-ability of humidity at some elevated layer). Maybe most of the cases studied here are CBL fog more than pure radiation fog (see Tardif and Rasmussen (2007) classification (http://journals.ametsoc.org/doi/pdf/10.1175/JAM2516.1)). Could the authors discuss or demonstrate a little bit more about this? Maybe the authors could check somehow if the studied fogs are preceded by other foggy periods or if this fact (previous foggy days) can iinfluence the height of Hmax, since it seems logical that Hmax could be higher if fog were observed the previous days.

SOME SPECIFIC COMMENTS

Page 3, line 4: airports, air traffic. . .

Page 3, line 6: reduced, possibly affecting. . .

Page 3, line 6: Hence, accurate. . . (please, revise the use of commas in the whole text).

Page 3, line 10: References in chronological order (please, revise in the whole text).

Page 5, line 13: 100 m (please, revise the space before "m" in the whole text).

Page 6, line 1: whose size ranges.

Page 7, line 7: "under cloud-free atmosphere".

Page 7, line 12: ". . . and 3 km (Dupont et al. (2015), where the low visibility. . .")

Page 7, line 11: QFOG can also be similar to the definition of mist by AMS?

Page 17, line 15: I would recommend to add: "This suggests that once fog droplets. . ." (since visibility measurements are not available at several heights to be absolutely sure).

---

## Referee Comment (RC2) · Anonymous Referee #2 · 26 Jul 2016

GENERAL COMMENTS

This manuscript presents a practical method for providing fog alerts from routine measurements that will be of major benefit to society. The study takes a pragmatic approach suitable for routine measurements, together with excellent analysis, and demonstrates applicability to real cases. The manuscript provides a clear methodology that can potentially be objectively applied to numerous sites across the globe. I believe this paper is ready for publication after a few very minor modifications.

The method understandably uses threshold values for some of the alerts. A quick discussion on how sensitive the method is to the choice of threshold values would be very useful; the authors already note that the method will probably require some tuning

at different locations. The method also assumes homogeneous aerosol properties; there may be locations where this assumption may not be so reliable, could the authors elaborate a little on the likely impact in terms of fog alerts?

Fog is a rare event, so I assume it is better to have some 'false alarms' rather than any 'misses'.

It should also be noted that this method requires ALCs that have full overlap already at quite low altitudes so that reliable attenuated backscatter values are available from 50 m or so in height. This is a major implication in terms of instrument selection.

SPECIFIC COMMENTS

Page 1, line 26: I suggest that you start this sentence with 'We find that an alert for pre-fog conditions predominantly occurs ..'

Page 2, line 1: Replace 'sensitive to relative humidity' with 'sensitive to the relative humidity'.

Page 3, line 3: I suggest that you replace 'formation of a liquid layer near the ground' with 'formation of a layer containing liquid water droplets near the ground'.

Page 3, lines 7-8: Replace 'air traffic in the' with 'air traffic over the'.

Page 3, line 21: Replace 'scores' with 'skill scores'.

Page 4, line 10: Here and elsewhere, replace 'could' with 'can'.

Page 5, line 3: Replace 'are' with 'have been'.

Page 5, line 10: Table 1 states 910 nm, as do the manuals. Also page 16, line 17.

Page 6, line 1: Replace 'size' with 'sizes'.

Page 6, lines 2,3: Suggest that you state 'relatively low SNR' as the latest CL51 instrument usually exhibits high SNR in the boundary layer at night.

Page 6, line 24: Suggest replacing 'for bi-axial ALCs full overlap can be reached' with 'for bi-axial A LCs full overlap can only be reached'.

Page 7, lines 9, 10, 11, 13 (and elsewhere): Remove the indefinite article 'a' when referring to fog. Also, do not use the plural 'fogs' on line 11.

Page 7, lines 7-9,11: Use 'vertically-developed' and 'quasi-radiation'.

Page 7, lines 11-12: I suggest that you combine these two sentences otherwise it could be read as saying that all fog is due to inactivated haze particles.

Page 8, lines 14, 15: The real part of the refractive index may be lower for pure water than aerosol; here you are dealing with a mixture of water and aerosol and it is the relative change in the amount of water in the droplet that changes the refractive index. Note that the imaginary part of the refractive index is small but not zero for pure water, and that this also varies with wavelength.

Page 9, line 12: Replace 'properties as' with 'properties such as'.

Page 9, line 14: Replace 'describe that' with 'assume that'.

Page 10, line 16: Replace 'follow' with 'follows'.

Page 10, line 18: Replace 'dividing' with 'by dividing'.

Page 10, line 20: Replace 'in periods' with 'over periods'. Is there a suitable reference showing the instrument calibration stability (i.e. that the calibration factor and the overlap function are stable)?

%%%Page 13, line 10: Replace 'had' with 'has'.

Page 16, line 9: Should this be attenuated backscatter?

Page 17, lines 16, 21 and elsewhere (page 21): Suggest using 'm hr-1' rather than 'm/h'.

Page 18, line 4: Replace 'alert' with 'alerts'.

Page 19, line 2; Replace 'moistest' with 'most moist'.

Page 22, line 11: Replace 'pre fog' with 'pre-fog'.

Table 1: Full optical overlap for CL51 much lower than the 500 m stated here. You could include temporal resolution in this table.

Figure 3: There is a slight departure from the fit between 75% and 85% which looks like hysteresis in the aerosol response to RH - is this during the increase in RH or decrease?

Figure 5: Replace 'time series' with 'Time series'.

Figures 6, 7: Note somewhere in the caption that white is at either end of the colour scale (noise and cloud), or fill cloud (> 1e-4 m-1 sr-1) with some colour.

Figures 6-9: In panel b, the colour for the line at the top of the panel (PARAFOG status?) is not described in the caption. What do you mean by 'Pre-fog conditions are clear'? Clear skies?

Figure 11: It would be easier to read this figure if the colour scale was separate from the plot panels.

---

## Referee Comment (RC3) · Anonymous Referee #3 · 27 Jul 2016

GENERAL COMMENTS

This paper describes a new algorithm for now-casting of fog formation. Fog now-casting can be very valuable, especially for airports where low visibility can cause major disturbances. This algorithm uses the hygroscopic growth of the attenuated backscatter to provide alerts prior the formation of radiation fog. The algorithm was tested on 45 fog cases from 2011 to 2015 at two sites (SIRTA and Uccle).

This manuscript presents a new and valuable technique that can be applied on a large number of existing stations. The manuscript is clearly written and the algorithm is well described. The analysis of the alert occurrence also provides interesting results concerning the altitude where the cooling process lead to aerosol activation.

[Figure]

However, the possibility of false-alarms was not evaluated. A discussion about the algorithm limitations is missing to fully appreciate the usefulness of these observations to complement the Numerical Weather Predictions (NWP).

Therefore I recommend the publication of this manuscript but only after the correction of the following major comment.

MAJOR COMMENT

As the author mentioned in the introduction, air traffic at busy airports can be significantly disrupted in case of low visibility. New observations and warnings can be very valuable for airport forecasters but only if their forecasting skills are higher than NWP's.

This study shows an evaluation of PARAFOG performance between 2011 and 2015. If no false alarm occurred during this period, the authors should mentioned it. If it is not the case, the authors should present a case study and statistics about the hit-rate and false-alarm occurrences. For the same SIRTA site, Menut et al. (2013) described a method to forecast fog from ground based measurements with an hit-rate of 87%. They also mentioned a false-alarm rate of 39%. Similar statistics are required to evaluate the performances of the PARAFOG algorithm.

Further discussions about algorithm limitations and possible improvements of the algorithm would also be valuable.

SPECIFIC COMMENTS

Page 3 line 14 and l 20: Please replace "Roman-Cascon et al. 2015" by "Roman-Cascon et al. 2016"

Page 5 lines 19-21: What is the impact over the oversampling on the PARAFOG algorithm ?

Page 11 line 8 eq. 12: Please define the greek letter xi

Page 15 line 10 eq. 18: What does the acronym RG stands for ?

Page 16 line 7: could you discuss the impact of these thresholds ?

Page 19 line 4: How the mean extinction was calculated ? What was the Lidar Ratio assumption?

Page 27 line 19 : Please replace " Román-Cascón, C., Steeneveld, G. J., Yagüe, C., Sastre, M., Arrillaga, J. A., & Maqueda, G., Forecasting radiation fog at climatologically contrasting sites: evaluation of statistical methods and WRF. Quarterly Journal of the Royal Meteorological Society, 2015" by "Román-Cascón, C., Steeneveld, G. J., Yagüe, C., Sastre, M., Arrillaga, J. A., & Maqueda, G., Forecasting radiation fog at climatologically contrasting sites: evaluation of statistical methods and WRF. Quarterly Journal of the Royal Meteorological Society, 2016"

Page 33 figure 1: Please add units for the relative difference.

Page 33 figure 1: Please check the caption. "(bottom)" and "(top)" are inverted

Page 37 figure 5: Please check the caption. Could you replace "time series" by "scatter plot".

Page 38 figure 6: Please check the caption. Units is missing for the altitude of aerosol activation ("100m agl" instead of "100 agl"). (a) and (b) labels are missing, could you replace it by (top) and (bottom)?

Page 39 figure 7: Please check the caption. Units is missing for the altitude of aerosol activation ("100m agl" instead of "100 agl"). (a) and (b) labels are missing, could you replace it by (top) and (bottom)?

Page 39 figure 7: Where is the horizontal visibility line?

Page 40 figure 8: (a) and (b) labels are missing, could you replace it by (top) and (bottom)?

Page 41 figure 9: (a) and (b) labels are missing, could you replace it by (top) and (bottom)?

---

## Author Comment (AC1) · 23 Sep 2016

amt-2016-182
Title : Radiation fog formation alerts using attenuated backscatter power from automatic Lidars and ceilometers

Response to Reviewer 1 comments

The authors would like to thank the Anonymous reviewer #1 for helpful comments and suggestions. Reviewer comments are shown in black colour. A response is provided for each comment and shown in blue colour.

Reviewer 1 introductory comment:
This paper presents a very interesting method for the nowcasting of radiation fog using measurements from Lidars and ceilometers, as well as some additional instrumentation. The findings are of great interest for the forecasting of radiation fog, especially in areas with availability of these instruments (e.g. airports, where the problems associated with fog are especially important). The paper is well structured, although other options for the structure of the paper are possible. A good introduction to the algorithm is presented, where the authors show clearly all the problems/solutions for the applicability of the method. Then, the authors present more specific analyses for four case studies, as well as several results for 45 cases of radiation fog at two locations. I strongly recommend the publication of this paper in Atmospheric Measurements Techniques journal, since it provides a clear, good, interesting and potentially applicable method for the nowcasting of radiation fog. However, I have also some minor comments that could be addressed before the final publication.

GENERAL COMMENTS

REV1GC1:  The Introduction section provides a good overview of the current problems in fog forecasting and the process of hygroscopic growth of fog condensation nuclei. There is a lack of references from lines 4 to 12 of page 4. I recommend to include more references here.

Response to REV1GC1: four references have been added to this paragraph.

REV1GC2: Section 2 introduces the sites, the differences between the instrumentation used at SIRTA and UCCLE. In this section, I think that section 2.3 (fog dataset description) is more a description of types of radiation fogs than a description of the cases used in the work, which is given in Table 3. Maybe the authors could re-consider the inclusion of this subsection in the final paper.

Response to REV1GC2: we agree with the reviewer that the section was ill named. We rename Section 2.3 as "Radiation fog types used in this study". The last sentence of Section 2.3 is changed to "Our study relies on 45 cases of developed radiation fog, as defined above, observed at SIRTA and UCCLE."

REV1GC3: Section 3 is the most theoretical part of the paper, where relationships of relative hu- midity, aerosol properties and measurements from ALCs are given. In this section, it is also explained technically how the hygroscopic growth of aerosols can be derived from

attenuated backscatter profiles of ALCs and RH measurements from a hygrometer. Since I am not an expert on this, I cannot correct or make suggestions on the development of these relationships and on the experiments performed. However, I have found this section very interesting and clear for readers not familiar with these relationships.

Response to REV1GC3: We thank the reviewer for the positive feedback on Section 3.

REV1GC4: In Section 4, an algorithm to detect rapid changes in aerosol backscatter is presented, which is used to detect pre-fog periods.

In step 1 of the algorithm (start), the authors could also suggest the use of previous works on detection of pre-fog periods to complement or improve this step, such those works presented by Menut et al. (2014) (DOI 10.1007/s10546-013-9875-1) or in Román-Cascón et al. (2016) (DOI: 10.1002/qj.2708). The results used in these works could be a good complement to improve this step, although more instrumentation would be required. I think the authors could suggest this in this part.

Briefly, all the steps and thresholds described and determined in the work seem to be quite studied and analysed by the authors, which demonstrate that the preparation of the work have been done in a proper way.

Response to REV1GC4: We agree with the reviewer that other meteorological parameters could be used to determine if PARAFOG should be turned ON or OFF. For example the authors cited above use also air temperature temporal gradient, wind speed and Net infrared flux. The temperature temporal gradient distribution in pre-fog condition reveals predominantly negative values. It could be added as criteria to turn PARAFOG ON. High wind speed could be added as criteria to turn PARAFOG OFF. We do not use net infrared fluxes directly, instead we use cloud fraction as a proxy to determine if radiative cooling conditions are present or not.

The following sentence is added at the bottom of page 13: "Other parameters could also be used to identify conditions that are potentially favorable for fog formation, such a air temperature temporal gradient and horizontal wind speed, as suggested by Menut et al. (2014), although more instrumentation would be required. "

REV1GC5: In subsection 4.1, the authors test the algorithm over 3 situations at SIRTA and 1 at UCCLE.
Figures 6, 7, 8, 9: I would recommend to include more details about the figures in the figure captions. For example: "Figure 6: Time series presenting measurements and alert levels in pre-fog conditions on 15-16 November 2011 at SIRTA. (a) ALC attenu- ated backscatter (colour contours) and horizontal visibility (grey line). (b) Alert levels (colours) and altitude of Hmax (purple points)."
- Figures 6, 7, 8, 9: Please, add height of measurement of horizontal visibility. Add also the meaning of horizontal green and orange lines in figure captions.
- Figure 7: Where is the line of visibility?

- Figure 8: "Low-level alert 90 min before fog formation. Severe alert 30 min before fog formation" (is this correct?)

Response to REV1GC5: Figure caption text has been modified following the suggestion. A sentence has been added to explain the meaning of the green/orange line. Visibility measurements are not available in UCCLE, hence they don't appear on Figure 7. Figure 8 caption has been changed.

REV1GC6: In the whole text, I think the term "low-level" (referring to alert level) can lead to confusion (alert in the low-level atmosphere). I recommend to use the term "minor" or another one to avoid this.

Response to REV1GC6: The word "minor" is a good suggestion instead of "low-level". Text and figures have been changed to reflect that suggestion.

REV1CG7: Case studies of section 4.2 and section 5.4 (alert altitudes): It is shown how in many cases of radiation fogs studied, the activation of the droplets occurs at certain height at both sites (around 100 m (mean) and even higher at the urban site) and then it is observed the subsidence of Hmax. For me, it is not clear if this also means the subsidence of the "fog layer", as written in page 19 (lines 24 to 26) or only the activation. If the results refer to the subsidence of the fog layer, I wonder if this could be checked with different visibilimeters at different heights (maybe not available) or if this result could be an artefact from the algorithm or from the ALC measurements (maybe with more problems with layers below 70-100 m (not only at UCCLE)). These results are quite interesting, since normally (and based on other studies), radiation fog normally starts to be formed quite close to the ground and then it evolves into higher heights, except for "cloud-base-lowering (CBL) fog types", when a fog occurring certain day is the result of the lowering of "elevated" clouds from previous fog of the precedent day or even the result of the dissipation of previous fog (but with more avail- ability of humidity at some elevated layer). Maybe most of the cases studied here are CBL fog more than pure radiation fog (see Tardif and Rasmussen (2007) classification (http://journals.ametsoc.org/doi/pdf/10.1175/JAM2516.1)). Could the authors discuss or demonstrate a little bit more about this? Maybe the authors could check somehow if the studied fogs are preceded by other foggy periods or if this fact (previous foggy days) can influence the height of Hmax, since it seems logical that Hmax could be higher if fog were observed the previous days.

Response to REV1GC7: We agree with the reviewer that typically a radiation fog occurs due to cooling of the air just above the surface. This occurs predominantly in the absence of a cloud ceiling. However, as noted by Tardif and Rasmussen (2007) in their fog definition, a radiation fog can also start with an elevated fog formation, with a cloud ceiling near or below 100 m followed shortly by fog at the surface. Here, prior to the elevated fog formation, the sky is predominantly cloud-free enabling surface radiative cooling. We find that those are quite frequent at both sites studied here.
In contrast a "cloud-base-lowering (CBL) fog type" corresponds to a gradual lowering of a cloud ceiling during several hours priors to fog occurrence at the surface. Hence during several hours, the sky is mostly overcast, preventing any significant surface radiative cooling.

SOME SPECIFIC COMMENTS
Page 3, line 4: airports, air traffic. . .
Page 3, line 6: reduced, possibly affecting. . .
Page 3, line 6: Hence, accurate. . . (please, revise the use of commas in the whole text).
Page 3, line 10: References in chronological order (please, revise in the whole text).
Page 5, line 13: 100 m (please, revise the space before "m" in the whole text).
Page 6, line 1: whose size ranges.
Page 7, line 7: "under cloud-free atmosphere".
Page 7, line 12: ". . . and 3 km (Dupont et al. (2015), where the low visibility. . .")
Page 7, line 11: QFOG can also be similar to the definition of mist by AMS?
Page 17, line 15: I would recommend to add: "This suggests that once fog droplets. . ." (since
visibility measurements are not available at several heights to be absolutely sure).

Response to specific comments: all 10 suggested changes have been made.

---

## Author Comment (AC2) · 23 Sep 2016

amt-2016-182
Title : Radiation fog formation alerts using attenuated backscatter power from automatic Lidars and ceilometers

Response to Reviewer 2 comments.

The authors would like to thank the Anonymous reviewer #2 for helpful comments and suggestions. Reviewer comments are shown in black colour. A response is provided for each comment and shown in blue colour.

Anonymous Referee #2

**GENERAL COMMENTS (GC)**
This manuscript presents a practical method for providing fog alerts from routine measurements that will be of major benefit to society. The study takes a pragmatic approach suitable for routine measurements, together with excellent analysis, and demonstrates applicability to real cases. The manuscript provides a clear methodology that can potentially be objectively applied to numerous sites across the globe. I believe this paper is ready for publication after a few very minor modifications.

REV2GC1: The method understandably uses threshold values for some of the alerts. A quick discussion on how sensitive the method is to the choice of threshold values would be very useful; the authors already note that the method will probably require some tuning at different locations. The method also assumes homogeneous aerosol properties; there may be locations where this assumption may not be so reliable, could the authors elaborate a little on the likely impact in terms of fog alerts?
Fog is a rare event, so I assume it is better to have some 'false alarms' rather than any 'misses'.

Response to GC1: Changing alert threshold values, and changing reference relative humidity both have an impact on the time at which the alert occurs. Changing the reference relative humidity from 40% to 80% delays a minor-level alert occurrence by 30-60 min. Changing the minor-level alert value by one order of magnitude also delays the alert occurrence by 60 min. This last sentence has been added to the text page 16, lines 13-14. Depending on the nature of the aerosols, hygroscopic growth may require different relative humidity conditions. A more in depth study involving more data is required to study alert occurrences depending the aerosol chemical nature or size distributions.

REV2GC2: It should also be noted that this method requires ALCs that have full overlap already at quite low altitudes so that reliable attenuated backscatter values are available from 50 m or so in height. This is a major implication in terms of instrument selection.

Response to GC2: Alerts at minor, moderate and severe levels are computed based on Eq. 18, that depend on the ratio of an ALC backscatter at a given time to that of an ALC backscatter in a drier condition that occurred earlier. Reaching full overlap is not required to derive this ratio, as the overlap value cancels out in the ratio. However, the overlap function could be temperature dependent and hence could change with time. Hence calculating the

backscatter ratio at heights where the overlap function is less than 0.1 could introduce significant uncertainties in the ratio calculation. For example, Vaisala does not recommend to use the CL51 data below 50 m agl. For the CL31, the lower limit is about 10 min.

**SPECIFIC COMMENTS**

Page 1, line 26: I suggest that you start this sentence with 'We find that an alert for pre-fog conditions predominantly occurs ..'

Response: changed.

Page 2, line 1: Replace 'sensitive to relative humidity' with 'sensitive to the relative humidity'.

Response: changed.

Page 3, line 3: I suggest that you replace 'formation of a liquid layer near the ground' with 'formation of a layer containing liquid water droplets near the ground'.

Response: changed.

Page 3, lines 7-8: Replace 'air traffic in the' with 'air traffic over the'. Page 3, line 21: Replace 'scores' with 'skill scores'.

Response: changed.

Page 4, line 10: Here and elsewhere, replace 'could' with 'can'. Page 5, line 3: Replace 'are' with 'have been'.

Response: changed.

Page 5, line 10: Table 1 states 910 nm, as do the manuals. Also page 16, line 17. Page 6, line 1: Replace 'size' with 'sizes'.

Response: changed.

Page 6, lines 2,3: Suggest that you state 'relatively low SNR' as the latest CL51 instrument usually exhibits high SNR in the boundary layer at night.

Response: changed.

Page 6, line 24: Suggest replacing 'for bi-axial ALCs full overlap can be reached' with 'for bi-axial A LCs full overlap can only be reached'.

Response: changed.

Page 7, lines 9, 10, 11, 13 (and elsewhere): Remove the indefinite article 'a' when referring to fog. Also, do not use the plural 'fogs' on line 11.

Response: changed.

Page 7, lines 7-9,11: Use 'vertically-developed' and 'quasi-radiation'.

Response: changed.

Page 7, lines 11-12: I suggest that you combine these two sentences otherwise it could be read as saying that all fog is due to inactivated haze particles.

Response: changed.

Page 8, lines 14, 15: The real part of the refractive index may be lower for pure water than aerosol; here you are dealing with a mixture of water and aerosol and it is the relative change in the amount of water in the droplet that changes the refractive index. Note that the imaginary part of the refractive index is small but not zero for pure water, and that this also varies with wavelength.

Response: changed to "The imaginary part of water is near zero…".

Page 9, line 12: Replace 'properties as' with 'properties such as'. Page 9, line 14: Replace 'describe that' with 'assume that'.

Response: changed.

Page 10, line 16: Replace 'follow' with 'follows'.

Response: changed.

Page 10, line 18: Replace 'dividing' with 'by dividing'.

Response: changed.

Page 10, line 20: Replace 'in periods' with 'over periods'. Is there a suitable reference showing the instrument calibration stability (i.e. that the calibration factor and the overlap function are stable)?

Response: there is no reference showing Vaisala ceilometer calibration stability. Studies performed in the framework of the TOPROF COST action show that calibration coefficient change over time on seasonal scales (not diurnal scales). Day-night differences of about 10-20% in the overlap function of Lufft CHM15k ceilometers have been found by Hervo et al. (2016). According to Kotthaus et al. (2016), overlap function uncertainties in CL31 ceilometers are expected to be less than 10% due to low internal temperature variations.

Hervo, M., Poltera, Y., and Haefele, A.: An empirical method to correct for temperature dependent variations in the overlap function of CHM15k ceilometers, Atmos. Meas. Tech., doi:10.5194/amt-2016-30, in press, 2016.

Kotthaus, S., O'Connor, E., Münkel, C., Charlton-Perez, C., Gabey, A. M., Grimmond, C. S. B., and Haeffelin M.: Recommendations for processing atmospheric attenuated backscatter profiles from Vaisala CL31 Ceilometers, Atmos. Meas. Tech., doi:10.5194/amt-2016-87, 2016.

Page 13, line 10: Replace 'had' with 'has'.

Response: changed.

Page 16, line 9: Should this be attenuated backscatter?

Response: changed to attenuated backscatter.

Page 17, lines 16, 21 and elsewhere (page 21): Suggest using 'm hr-1' rather than 'm/h'.

Response: changed to 'm h$^{-1}$' page 17 and 21

Page 18, line 4: Replace 'alert' with 'alerts'.

Response: changed.

Page 19, line 2; Replace 'moistest' with 'most moist'. Page 22, line 11: Replace 'pre fog' with 'pre-fog'.

Response: changed Page 19. Page 22 'pre radiation fog' replaced by 'pre-radiation-fog'

Table 1: Full optical overlap for CL51 much lower than the 500 m stated here. You could include temporal resolution in this table.

Response: Wiegner et al. 2014 suggest that the CL51 signal is generally overestimated below 500 m due to an internal overlap correction function applied by the VAISALA software. Unfortunately this function is unknown to the user. During the Ceilinex campaign http://ceilinex2015.de/special-topics/overlap, a comparison between two CL51 on the same site showed significant differences in the CL51 signal below 500 m probably due to this internal overlap correction function. The figure caption now refers to this publication for further details about the CL51 optical overlap function.

Wiegner, M., Madonna, F., Binietoglou, I., Forkel, R., Gasteiger, J., Geiß, A., Pappalardo, G., Schäfer, K., and Thomas, W.: What is the benefit of ceilometers for aerosol remote sensing? An answer from EARLINET, Atmos. Meas. Tech., 7, 1979-1997, doi:10.5194/amt-7-1979-2014, 2014.

Temporal resolution added to Table 1.

Figure 3: There is a slight departure from the fit between 75% and 85% which looks like hysteresis in the aerosol response to RH - is this during the increase in RH or decrease?

Response: Figure 3 shows data for a monotonous increase of RH (no decrease, except a plateau near 65% RH), so the departure from the fit cannot be explained by a hysteresis in the aerosol response to RH.

Figure 5: Replace 'time series' with 'Time series'.

Response: changed.

Figures 6, 7: Note somewhere in the caption that white is at either end of the colour scale (noise and cloud), or fill cloud (> 1e-4 m-1 sr-1) with some colour.

Response: RCS values below 1e-7 m-1 sr-1 are now shown in black instead of white.

Figures 6-9: In panel b, the colour for the line at the top of the panel (PARAFOG status?) is not described in the caption. What do you mean by 'Pre-fog conditions are clear'? Clear skies?

Response: PARAFOG status explained in the caption now. 'Pre-fog conditions are clear' changed to 'Pre-fog conditions are cloud-free'.

Figure 11: It would be easier to read this figure if the colour scale was separate from the plot panels.

Response: changed.

---

## Author Comment (AC3) · 23 Sep 2016

amt-2016-182
Title : Radiation fog formation alerts using attenuated backscatter power from automatic Lidars and ceilometers

Response to Reviewer 3.

The authors would like to thank the Anonymous reviewer #3 for helpful comments and suggestions. A response is provided for each comment, shown in blue color.

Reviewer 3 introductory comment:

Anonymous Referee #3

GENERAL COMMENTS

This paper describes a new algorithm for now-casting of fog formation. Fog now- casting can be very valuable, especially for airports where low visibility can cause major disturbances. This algorithm uses the hygroscopic growth of the attenuated backscatter to provide alerts prior the formation of radiation fog. The algorithm was tested on 45 fog cases from 2011 to 2015 at two sites (SIRTA and Uccle).

This manuscript presents a new and valuable technique that can be applied on a large number of existing stations. The manuscript is clearly written and the algorithm is well described. The analysis of the alert occurrence also provides interesting results concerning the altitude where the cooling process lead to aerosol activation.

However, the possibility of false-alarms was not evaluated. A discussion about the algorithm limitations is missing to fully appreciate the usefulness of these observations to complement the Numerical Weather Predictions (NWP).

Therefore I recommend the publication of this manuscript but only after the correction of the following major comment.

MAJOR COMMENT

As the author mentioned in the introduction, air traffic at busy airports can be significantly disrupted in case of low visibility. New observations and warnings can be very valuable for airport forecasters but only if their forecasting skills are higher than NWP's.

This study shows an evaluation of PARAFOG performance between 2011 and 2015. If no false alarm occurred during this period, the authors should mentioned it. If it is not the case, the authors should present a case study and statistics about the hit-rate and false-alarm occurrences. For the same SIRTA site, Menut et al. (2013) described a method to forecast fog from ground based measurements with an hit-rate of 87%. They also mentioned a false-alarm rate of 39%. Similar statistics are required to evaluate the performances of the PARAFOG algorithm.

Further discussions about algorithm limitations and possible improvements of the algorithm would also be valuable.

Response to Major Comment:

This is a very important issue, as in the end, one wants to know how often a fog prediction method provides "hits" and "false alarms" as this is a common metrics. In our paper, we present the basis for the study of fog formation by means of the light scattering measured by ALCs and we present PARAFOG to derive pre-fog formation alerts in order to highlight the potential of this methodology. Therefore, we discuss the occurrences and characteristics of these alerts based on 45 fog case studies (fog occurs in each case). We also demonstrate that the PARAFOG algorithm can be used on two different ALC datasets measured at two different sites. These results should be considered as the basis for the development of an algorithm for nowcasting fog formation.

Our work should be evaluated according to i) the set of observations in near real-time that could be useful to track the evolution of key processes and key parameters that drive fog formation and ii) the experimental observations that could complement the information predicted by NWP models that is made available to airport forecasters in support of their fog forecast.

From our recent experience participating in a field campaign at Charles-de-Gaulle airport, it is our understanding that airport forecasters use an ensemble of information (several NWP forecasts, satellite measurements, surface measurements, knowledge of local climatology) to derive their own "subjective" fog forecast. In our paper, we suggest that there is useful information in the ALC attenuated backscatter time series.

To address the question of "hit rates" and "false alarm rates", we must develop a method to define what constitutes a "hit" and a "false alarm" based on PARAFOG alert levels. We must study how alerts should be interpreted to make an alarm in an objective way, which will allow us to derive hit rates and false alarms rates based on the occurrence of low visibility. This requires an in-depth study, preferably at multiple sites to provide a robust evaluation. We are currently preparing a follow-on study based on ALC measurements at several locations in Europe.

To reflect this discussion, the following changes have been made in the manuscript:

Last paragraph of Section 4 (page 18), the following sentence has been added: "The behavior of the PARAFOG algorithm prior to quasi-fog situations should be tested further to estimate the potential for minor, moderate and severe alerts to occur in such conditions."

First paragraph of Section 5 (page 18), the text within parenthesis has been added: "…based on about 45 fog cases (fog occurs in each case) observed near Paris and Brussels…"

Last paragraph of the conclusion (page 23) has been rewritten: "To further evaluate the performance of PARAFOG, several developments are suggested: (1) a method for objective interpretation of alert levels should be developed to derive hit rates and false alarm rates; (2) performance tests should be carried out at other locations using datasets that include both pre-fog events and non-fog events; (3) alert threshold values should be adapted to reference relative humidity, and possibly to aerosol types using for example PM2.5 measurements."

We agree with the referee that the limitations of the methodology should be discussed. The following paragraph has been added after the fourth paragraph of the conclusions (page 23):

"Known limitations in our ability to track hygroscopic growth of aerosols using PARAFOG are (1) the minimum height at which ALC measurements can be reliably used due to ALC optical overlap; (2) water vapor absorption at 905-910 nm that affects attenuated backscatter values as specific humidity changes; (3) change in aerosol type (e.g., form marine salt to anthropogenic aerosols) within a few hours prior to fog formation."

**SPECIFIC COMMENTS**

Page 3 line 14 and l 20: Please replace "Roman-Cascon et al. 2015" by "Roman- Cascon et al. 2016"
Response: changed.

Page 5 lines 19-21: What is the impact over the oversampling on the PARAFOG algorithm ?

The major impact of the oversampling on PARAFOG is to delay up to 10 minutes (maximum) the determination when conditions are favorable and not favorable for pre-fog alerts based on the relative humidity (oversampled). In the worst cases, it may reduce the alert duration up to 10 minutes.

Page 11 line 8 eq. 12: Please define the greek letter xi

Response: We have changed the letter by 'z'. A sentence has been included in the text with the meaning:

"…where z is the variable representing altitude."

Page 15 line 10 eq. 18: What does the acronym RG stands for ?

Response: RG stands for Ration Gradient (or attenuated backscatter ration gradient). The definition of RG has been added to the text.

Page 16 line 7: could you discuss the impact of these thresholds ?

Response: these thresholds define the alerts (minor, moderate, and severe). Lower threshold values would alert about hygroscopic growths without leading to a real fog formation (false alarms). Conversely, larger threshold values would alert once the fog formation is almost finished reducing the prediction time.

Page 19 line 4: How the mean extinction was calculated ? What was the Lidar Ratio assumption?

Response: We make a rough estimate of the extinction, based on the two-way attenuation $T^2$ between the surface and 250-350 m above surface, assuming that both backscatter and

extinction coefficients are invariant with altitude in that range. A $T^2$ value of 0.95 in 250 m yields an extinction of ~0.85 x $10^{-04}$ $m^{-1}$ (Figure 10b), while a $T^2$ value of 0.40 in 350 m yields an extinction of ~1 x $10^{-03}$ $m^{-1}$ (Figure 10a). We do not make any Lidar ratio assumptions here. To reflect that this is a simple calculation, we change the extinction values reported in the text to "approximately $10^{-04}$ $m^{-1}$" and "approximately $10^{-03}$ $m^{-1}$"

Page 27 line 19 : Please replace " Román-Cascón, C., Steeneveld, G. J., Yagüe, C., Sastre, M., Arrillaga, J. A., & Maqueda, G., Forecasting radiation fog at climatologically contrasting sites: evaluation of statistical methods and WRF. Quarterly Journal of the Royal Meteorological Society, 2015" by "Román-Cascón, C., Steeneveld, G. J., Yagüe, C., Sastre, M., Arrillaga, J. A., & Maqueda, G., Forecasting radiation fog at climatologically contrasting sites: evaluation of statistical methods and WRF. Quarterly Journal of the Royal Meteorological Society, 2016"

Response: replaced.

Page 33 figure 1: Please add units for the relative difference.

Response: done, figure updated.

Page 33 figure 1: Please check the caption. "(bottom)" and "(top)" are inverted

Response: caption has been corrected.

Page 37 figure 5: Please check the caption. Could you replace "time series" by "scatter plot".

Response: figure caption changed.

Page 38 figure 6: Please check the caption. Units is missing for the altitude of aerosol activation ("100m agl" instead of "100 agl"). (a) and (b) labels are missing, could you replace it by (top) and (bottom)?

Response: caption corrected. (a) and (b) labels have been added.

Page 39 figure 7: Please check the caption. Units is missing for the altitude of aerosol activation ("100m agl" instead of "100 agl"). (a) and (b) labels are missing, could you replace it by (top) and (bottom)?

Response: caption corrected. (a) and (b) labels have been added.

Page 39 figure 7: Where is the horizontal visibility line?

Response: visibility measurements not available at UCCLE (caption updated accordingly).

Page 40 figure 8: (a) and (b) labels are missing, could you replace it by (top) and (bottom)?

Response: (a) and (b) labels have been added.

Page 41 figure 9: (a) and (b) labels are missing, could you replace it by (top) and (bottom)?

Response: (a) and (b) labels have been added.